# Thrombospondin-3 augments injury-induced cardiomyopathy by intracellular integrin inhibition and sarcolemmal instability

Tobias G. Schips[1], Davy Vanhoutte [1], Alexander Vo[1], Robert N. Correll[1], Matthew J. Brody[1], Hadi Khalil[1], Jason Karch[1,2], Andoria Tjondrokoesoemo[1], Michelle A. Sargent[1], Marjorie Maillet[1], Robert S. Ross [3] & Jeffery D. Molkentin [1,2]

Thrombospondins (Thbs) are a family of five secreted matricellular glycoproteins in verte-brates that broadly affect cell-matrix interaction. While Thbs4 is known to protect striated muscle from disease by enhancing sarcolemmal stability through increased integrin and dystroglycan attachment complexes, here we show that Thbs3 antithetically promotes sar-colemmal destabilization by reducing integrin function, augmenting disease-induced decompensation. Deletion of Thbs3 in mice enhances integrin membrane expression and membrane stability, protecting the heart from disease stimuli. Transgene-mediated over-expression of α7β1D integrin in the heart ameliorates the disease predisposing effects of Thbs3 by augmenting sarcolemmal stability. Mechanistically, we show that mutating Thbs3 to contain the conserved RGD integrin binding domain normally found in Thbs4 and Thbs5 now rescues the defective expression of integrins on the sarcolemma. Thus, Thbs proteins mediate the intracellular processing of integrin plasma membrane attachment complexes to regulate the dynamics of cellular remodeling and membrane stability.

[1] Department of Pediatrics, Cincinnati Children's Hospital Medical Center, Cincinnati, OH 45229, USA. [2] Howard Hughes Medical Institute, Cincinnati Children's Hospital Medical Center, Cincinnati, OH 45229, USA. [3] Division of Cardiology, Department of Medicine, University of California at San Diego School of Medicine, La Jolla, CA 92093, USA. Correspondence and requests for materials should be addressed to J.D.M. (email: jeff.molkentin@cchmc.org)

The connection between cells and the extracellular matrix (ECM) is of fundamental importance for normal organ function, which is especially germane to the heart given the need to transduce contractile forces and to maintain myocyte alignment for controlled rhythmicity[1]. Integrins are a primary structural linkage between the cellular cytoskeleton and the extracellular environment that is critical for cardiac homeostasis[2,3]. The integrin family consists of greater than 18 α-subunits and 8 β-subunits in mammals, forming at least 24 distinct combinations of covalently bound heterodimeric structural attachment moieties[4]. ECM constituents like collagens, laminins, and fibronectin are directly bound by the protruding extracellular domains of integrins that then link through focal adhesion protein complexes to the underlying cytoskeleton[4,5]. Integrin activity, avidity and affinity for ECM ligands can be modulated from the inside of the cell by post-translational modifications or by an array of accessory binding proteins[6]. Outside the cell a variety of matricellular proteins can also engage select integrin heterodimers[7].

In vertebrates the thrombospondin (Thbs) family consists of 5 homologous genes (Thbs1–5), of which Thbs1 and 2 form trimers while Thbs3, 4, and 5 form pentamers[8]. Thbs proteins are selectively expressed during embryonic development but in adult tissues their expression is essentially absent until an injury event occurs, where after all 5 family members are potentially induced[9,10]. Protective and detrimental effects have been associated with expression of Thbs proteins in certain cell types and at specific stages after injury[8]. Mice lacking each of the single Thbs family member genes are viable, as are combinatorial Thbs1/2 and Thbs1/3/5 targeted mice[11,12].

Very little is known concerning the role of Thbs3 during tissue homeostasis and disease except for a reported role during bone development[13]. Thbs3 is highly similar in protein sequence and overall structure to Thbs4 and both proteins are expressed in the heart[14,15] and upregulated during cardiac diseases[9,16]. The function of Thbs4 in heart and skeletal muscle has been investigated and shown to provide protection from multiple forms of disease simulated conditions or muscular dystrophy[17–21].

Thbs proteins have been linked to the endoplasmic reticulum (ER)-stress response. The ER-stress response reduces misfolded protein accumulation through the action of ER resident chaperones such as binding immunoglobulin protein (BiP), protein disulfide isomerase (PDI), calreticulin (CRT) and calnexin (CNX)[22]. The ER-stress response involves activation of 3 distinct branches consisting of PKR-like ER kinase, inositol requiring enzyme 1α (Ire1α) and activating transcription factor 6α (ATF6α)[23]. During their maturation in the secretory pathway, Thbs proteins interact with ER resident effectors such as BiP and ATF6α[24], resulting in the induction of an adaptive stress response, vesicular expansion, and reduced protein aggregation with augmented protein secretion[19,21,24]. In skeletal muscle, overexpression of Thbs4 dramatically enhanced sarcolemmal stability that rescued mouse models of muscular dystrophy through a mechanism involving enhanced integrin and dystroglycan-sarcoglycan attachment complexes in the plasma membrane[21].

Here we investigated the role of Thbs3 in the heart given the reliance of this organ on sarcolemmal stability for its homeostasis. Transgenic mice overexpressing Thbs3 specifically in cardiomyocytes showed exacerbated cardiac pathology with stress stimulation, while mice lacking Thbs3 were protected, which is the complete opposite of how Thbs4 functions when overexpressed or deleted in the heart or skeletal muscle[19–21,24]. Mice lacking Thbs1/2/4/5 that only have Thbs3 remaining were also generated and shown to have exacerbated pathology with cardiac stress stimulation, while Thbs1/2/3/4/5 quintuple null mice showed a rescue of this enhanced pathologic phenotype. Mechanistically we show that Thbs3 reduces integrin trafficking to the sarcolemma in the heart, while Thbs4 is cardioprotective by augmenting membrane residence of select integrins and β-dystroglycan of the glycoprotein attachment complex.

## Results

**Thbs3 is expressed in the diseased heart during ER stress.** While Thbs3 is not expressed in the adult mouse heart, induction of hypertrophy by cardiomyocyte-specific overexpression of activated calcineurin (CnA)[25] or 2 weeks of transverse aortic constriction (TAC) showed increased levels of Thbs3 protein (Fig. 1a and Supplementary Fig 1a). Thbs3 was also mildly but significantly induced in the hearts of $Csrp3^{-/-}$ mice[26], a model of dilated cardiomyopathy (Fig. 1a and Supplementary Fig 1a). Cardiac pathology in these select mouse models was also associated with ER stress and BiP and ATF6α protein induction (Supplementary Fig. 1a). A time course of TAC showed a trend towards induction of Thbs3 mRNA at 3 and 7 days but with significant and robust induction by 2 and 12 weeks (Supplementary Fig 1b). Immunohistochemical analysis of heart sections from CnA transgenic mice and mice subjected to TAC showed cardiomyocyte restricted, intracellular localization of Thbs3 protein with cytoplasmic desmin staining, which was directly coincident with the ER chaperone PDI (Fig. 1b, c). We also used a Thbs3 gene-targeted mouse line that contained a β-galactosidase (βgal) cDNA cassette to serve as a highly sensitive reporter of Thbs3 locus expression. While uninjured sham-operated mice showed no Thbs3 locus activation (βgal) in any cardiac cell-type from heart histological sections, 2 weeks of TAC stimulation produced abundant expression that was within cardiac myocytes (desmin positive), blood vessels (isolectin positive) and fibroblasts (vimentin positive), indicating that Thbs3 is ubiquitously expressed in an inducible manner under stress conditions in three major cardiac cell-types (Supplementary Fig 1c).

To model the induction of Thbs3 in diseased hearts and to analyze its function we generated transgenic mice overexpressing Thbs3 using an inducible cardiomyocyte-specific, tetracycline-regulated bi-transgenic system regulated by the α-myosin heavy chain (α-MHC) promoter[27] (Fig. 1d). In all experiments these double transgenic mice (DTG) were compared with animals only expressing the tetracycline transactivator (tTA). Thbs3 DTG hearts showed expression of Thbs3 and induction of the nuclear form of ATF6α, as well as multiple ER chaperones and transcriptional targets of ATF6α such as Armet, BiP and CRT (Fig. 1e). ER chaperones were upregulated to similar levels in hearts from Thbs3 DTG and Thbs4 DTG mice (See below, Supplementary Fig. 2a), consistent with our previous description of Thbs4 heart-specific and skeletal muscle-specific transgenic mice[19,21,24]. Also similar to past results with Thbs4, immunoprecipitation experiments revealed that Thbs3 could directly interact with ATF6α in neonatal rat ventricular cardiomyocytes (NRVM) overexpressing Thbs3 (βgal was the control) and an ER-retained mutant of ATF6α (Fig. 1f). Similar to what was observed in diseased hearts, overexpression of Thbs3 in transgenic mice again showed intracellular localization that was coincident with PDI in the ER compartment (Fig. 1g). Finally, ultrastructural analysis showed strong expansion of the ER-vesicular compartment with attached ribosomes (Fig. 1h), similar to what was observed in hearts and skeletal muscle overexpressing Thbs4[19,21], with no ultrastructural defects in mitochondria or sarcomeres (Supplementary Fig 1d).

Transgene-mediated Thbs3 overexpression and induction of ER-stress did not result in a baseline phenotype in the heart of these mice. Histological analysis by hematoxylin and eosin (H&E)

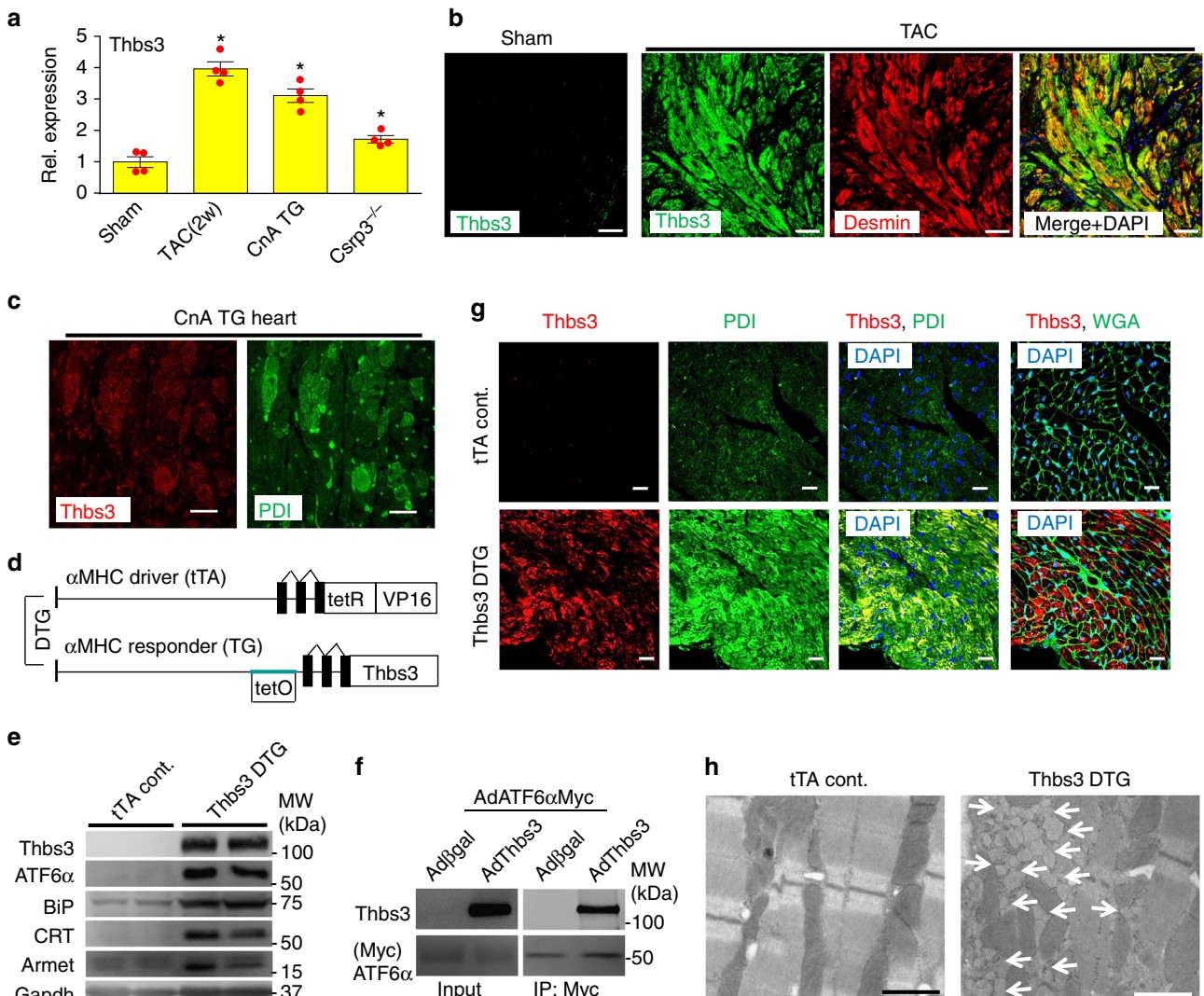

**Fig. 1** Thbs3 is expressed in the diseased heart and induces an adaptive ER stress response. **a** Quantitation of relative protein expression from Western blots for Thbs3 from heart tissue of mice that were sham-operated, subjected to 2 weeks of TAC, that contain the activated calcineurin A transgene (CnA) or that lack the *Csrp3* gene. *$P < 0.05$ vs sham by one-way ANOVA and Turkey multiple comparisons test. Results are from four experiments and error bars represent $+/-$SEM. **b** Immunohistochemistry for Thbs3 protein (green), desmin (red) and both merged with DAPI (blue) sham or TAC-operated hearts 12 weeks later. Scale bars are 10 μm. **c** Representative immunohistochemistry for Thbs3 (red) and PDI (green) from CnA transgenic hearts at 8 weeks of age. Scale bars are 10 μm. **d** Schematic diagram depicting the inducible bi-transgenic system regulated by tetracycline for inducible overexpression of Thbs3 in the heart. **e** Representative Western blots for Thbs3, the nuclear form of ATF6α, Armet, BiP, calreticulin (CRT) and Gapdh from hearts of tTA control and Thbs3 DTG mice. **f** Western blots for Thbs3 from neonatal rat ventricular myocytes (NRVM) infected with recombinant adenovirus expressing Myc-tagged ATF6α and βgal or Thbs3. Input control and immunoprecipitation (IP) of Thbs3 with Myc-tagged ATF6α are shown. **g** Representative immunohistochemistry for Thbs3 (red), WGA (green) or PDI (green) and DAPI (blue) from hearts of tTA control and Thbs3 DTG mice. Scale bars are 10 μm. **h** Transmission electron microscopy (EM) of heart sections from tTA control and Thbs3 DTG mice. The white arrows show expanded ER and vesicles only in Thbs3 DTG hearts. Scale bars are 1 μm

and Masson's trichrome staining failed to show histopathology or fibrosis of the heart when mice were analyzed up to 10 months of age (Supplementary Fig. 2b, c). Hearts from aged transgenic and control animals were analyzed by echocardiography and no differences were detected in fractional shortening (FS) (Supplementary Fig. 2d), cardiac dimensions during diastole and systole (Supplementary Fig. 2e) or heart weight-to-body weight (HW/BW) ratios (Supplementary Fig. 2f). There were also no changes in the calcium transient or sarcoplasmic reticulum calcium load in adult myocytes isolated from Thbs3 DTG hearts versus tTA controls (Supplementary Fig. 2g-k).

**Cardiac Thbs3 expression increases injury-induced disease.** Like Thbs3, cardiomyocyte-specific overexpression of Thbs4 in mice had no effect at baseline, yet Thbs4 transgenic mice were protected from injury-induced cardiac disease states[19]. Hence, we subjected 8 week-old Thbs3 DTG and tTA control animals to cardiac pressure overload by TAC stimulation for 12 weeks, which unexpectedly showed significantly greater hypertrophy with Thbs3 overexpression (Fig. 2a, b). Echocardiographic analysis showed that Thbs3 also significantly exacerbated ventricular remodeling and dilation with a greater loss in cardiac ventricular performance, lung edema and greater cardiac fibrosis compared

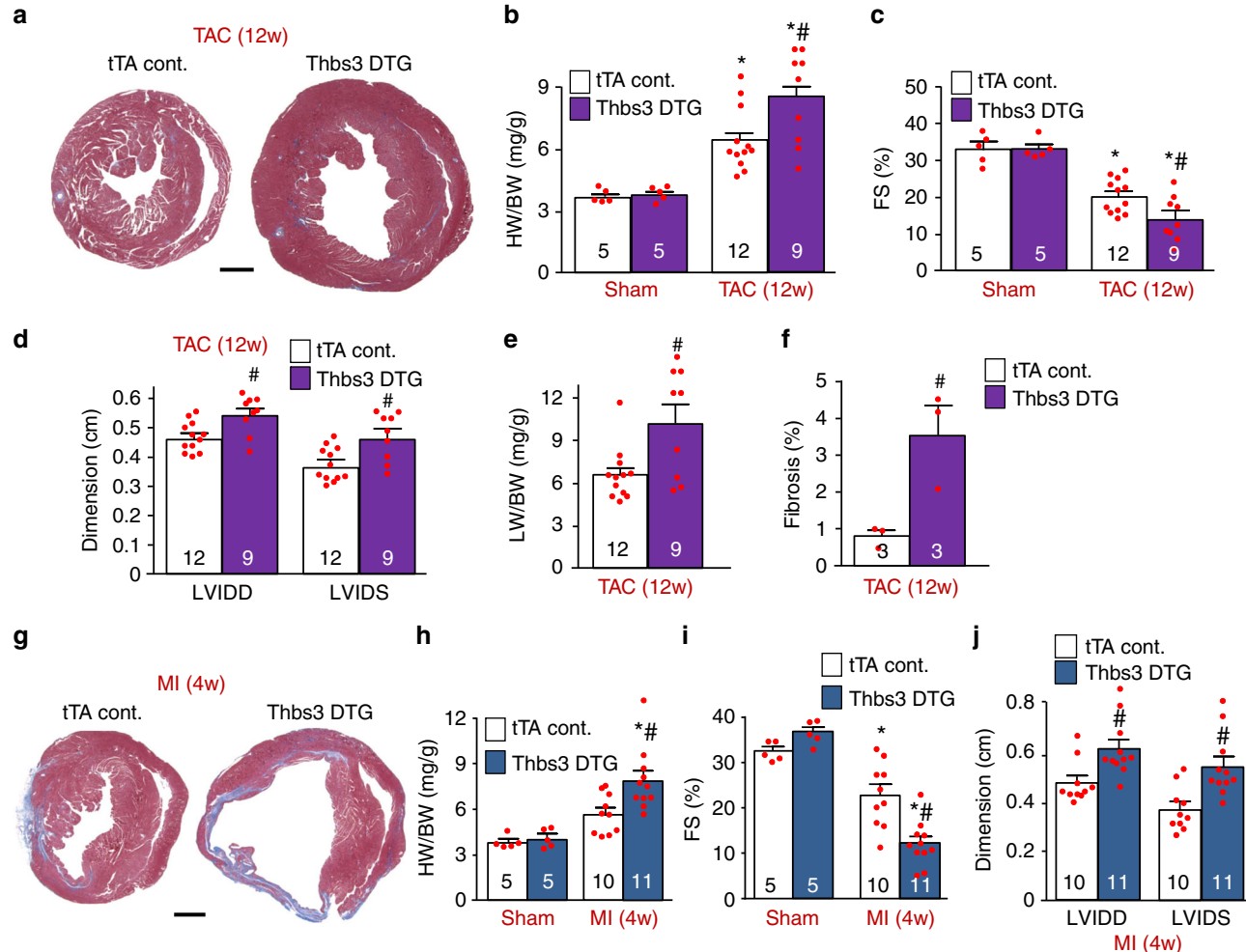

**Fig. 2** Cardiac-specific Thbs3 overexpression increases pathology after injury. **a** Low magnification images of tTA control and Thbs3 DTG cardiac histological sections stained with Masson's trichrome 12 weeks after TAC surgery. Scale bar is 1 mm. **b** Heart weight-to-body weight (HW/BW) ratio 12 weeks after TAC or sham surgery. **c** Echocardiography measured fractional shortening (FS) percentage and **d** left ventricular dimensions in diastole (LVIDD) and systole (LVIDS). **e** Pulmonary edema was analyzed by lung weight-to-body weight (LW/BW) ratios in the indicated groups of mice. **f** Percentage of fibrotic area was quantified from heart sections stained with Masson's trichrome. **g** Low magnification histological images of tTA control and Thbs3 DTG hearts, 4 weeks after MI surgery stained with Masson's trichrome. Scale bar is 1 mm. **h** HW/BW ratio 4 weeks after MI or Sham surgery in the indicated groups of mice. **i** Fractional shortening (FS) percentage and **j** LVIDD and LVIDS measured by echocardiography. Numbers of animals analyzed in each group for each procedure is indicated on the histograms. *$P < 0.05$ versus tTA control sham; #$P < 0.05$ versus tTA TAC or MI. Statistical analysis was performed using one-way ANOVA and Turkey multiple comparisons test. Error bars are $+/-$ standard error of the mean and number of mice used in each experiment are shown in the graphs

with controls (Fig. 2c–f). A similar augmentation in disease was observed when Thbs3 DTG animals were subjected to myocardial infarction (MI) injury over 4 weeks, showing increased heart size (Fig. 2g, h) and a greater loss in ventricular performance with enhanced ventricular dilation compared with tTA controls (Fig. 2i, j). These results suggest that Thbs3 is a maladaptive effector in the heart during disease stimulation, in contrast to Thbs4 that is cardioprotective[19].

**Thbs3 reduces surface integrin and sarcolemma stability.** Previous analysis of Thbs4 transgenic mice skeletal muscle showed a dramatic re-stabilization of the weakened sarcolemma in dystrophic mice through augmented membrane attachment complexes[21]. Moreover, previous studies have established a connection between Thbs and integrin proteins[28–30]. Given the opposing results of Thbs3 and Thbs4 after cardiac injury we performed a detailed analysis of integrin complexes in the cardiac

sarcolemma from the hearts of these 2 mouse models. Remarkably, Thbs3 overexpression lead to a uniform profile of inhibited integrin subunit expression at the sarcolemma, including α4, 5, 6, 7, 9, 10, and β1D integrin without changing proteins of the dystroglycan complex (Fig. 3a, Supplementary Fig 3a–k). By comparison, overexpression of Thbs4 had no inhibitory effect on α4, 5, 6, 7, 9, 10 integrins but instead increased the sarcolemmal content of integrin α2 and β3, along with increased β-dystroglycan levels (Fig. 3a and Supplementary Fig 3a-k). Importantly, immunohistochemistry for β1D, α5, or α7 integrins from Thbs3 DTG hearts showed reduced membrane signal versus tTA control hearts (Supplementary Fig 3l-n), although western blotting from total cellular extracts showed equal levels between control and Thbs3 DTG hearts. We next immunoprecipitated intact β1 integrin containing microsomal fractions from Thbs3 overexpressing NRVMs to analyze if Thbs3 protein was contained or excluded from this fraction of vesicles. While the input contained abundant Thbs3 and β1D integrin, after

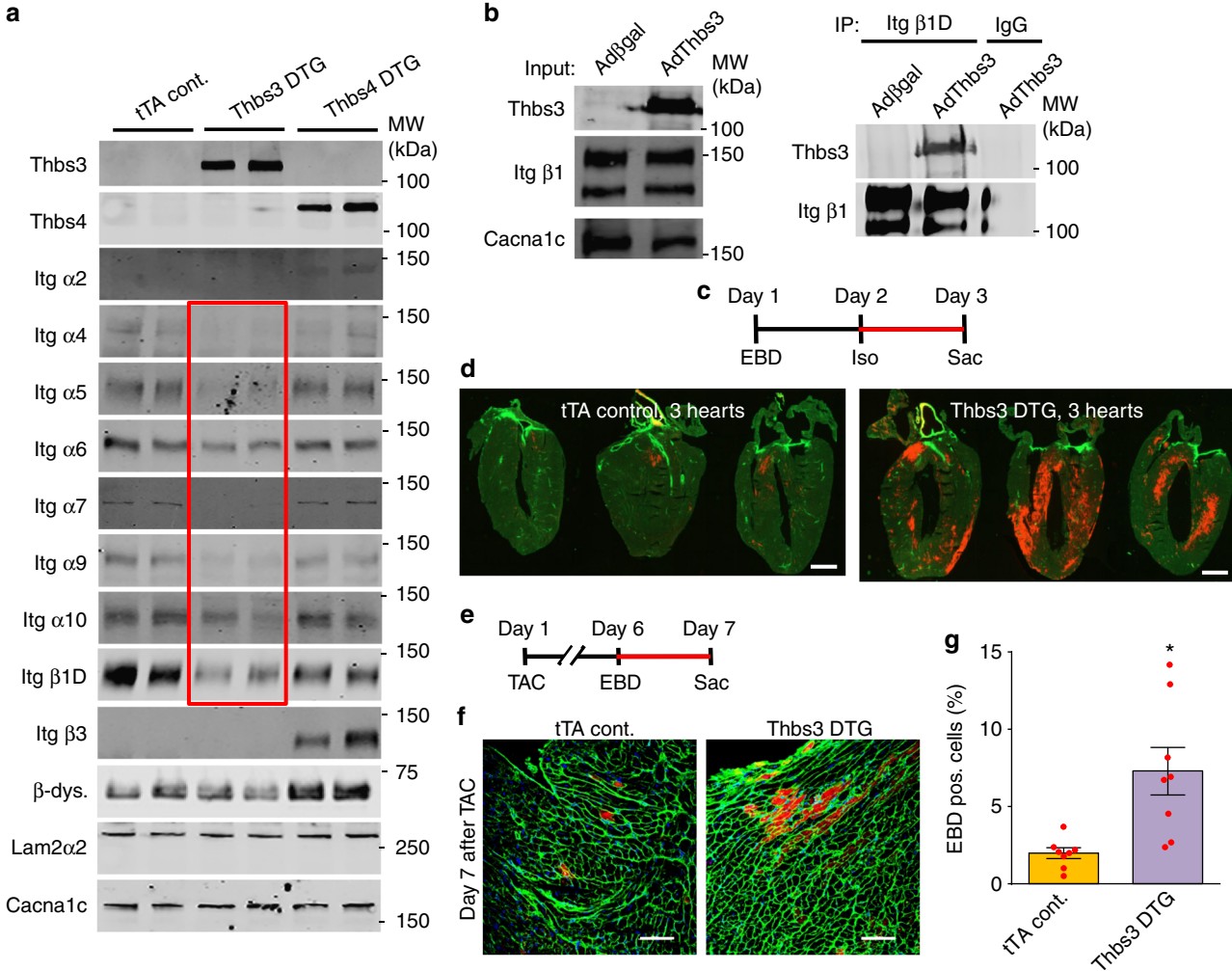

**Fig. 3** Thbs3 reduces surface integrin levels and destabilizes the sarcolemma. **a** Representative Western blots of sarcolemma protein extracts from hearts of tTA control, Thbs3 DTG, and Thbs4 DTG mice, assayed for integrin proteins and β-dystroglycan. Laminin2α2 and Cacna1c serve as loading controls. Quantitation is shown in Supplementary Figure 3a-k. The red box shows integrin proteins specifically reduced on the sarcolemma by Thbs3. **b** Western blots for input of microsomal protein extracts before immunoprecipitation for Thbs3 and β1D integrin and Cacna1c from NRVMs infected with Adβgal or AdThbs3 (left panel). The microsomal fraction was then immunoprecipitated (IP) for β1 integrin and blotted for the indicated proteins (right panel). Mouse IgG was used as a control for the IP. **c** Regimen of Evans blue dye (EBD) and isoproterenol (Iso, 300 mg/kg) injection to measure membrane permeability in the hearts of mice. **d** Representative histological heart sections from 3 tTA control and 3 Thbs3 DTG mice, imaged for WGA (green fluorescence) and leakage of EBD into the cardiomyocytes by red fluorescence subjected to the experimental regimen shown in **c**. Scale bars are 1 mm. **e** Regimen of Evans blue dye (EBD) injection after TAC surgery to measure membrane permeability in adult mice. Mice were sacrificed (Sac) 24 h after EBD injection, 1 week after TAC surgery. **f**, **g** Representative cardiac histological images of tTA control and Thbs3 DTG hearts 1 week after TAC surgery stained with WGA (green) and intracellular leakage of EBD (red) according to the experimental scheme shown in **e**, while **g** shows quantitation of EBD positive cells in the hearts of these mice. Scale bars are 100 μm. *$P < 0.05$ versus tTA control sham; two-tailed students $T$-test. Data are represented as percentage EBD positive cells ($\geq$3000 cells from $\geq$8 animals). Error bars are $+/-$ standard error of the mean and number of mice used is shown in the graph as individual data points

immunoprecipitation we observed that Thbs3 was indeed within the β1D integrin vesicular fraction, while a control for IgG immunoprecipitation showed no Thbs3 (Fig. 3b).

To characterize the effects of reduced integrin levels on cardiac sarcolemma stability we utilized an Evans blue dye (EBD) uptake assay to detect plasma membrane ruptures[31]. One bolus of isoproterenol (Iso) (Fig. 3c) or short-term TAC (see below, Fig. 3e) revealed significantly increased EBD cardiomyocyte membrane rupture events in hearts of Thbs3 DTG mice compared to control mice (Fig. 3d, f, g). These data are also consistent with the effect of decreased levels of integrins in the heart as predisposing to pathological remodeling[32], as well as heart disease observed in cardiomyocyte-specific transgenic mice

expressing a secretion incompetent mutant of Thbs4 that were characterized by reduced levels of dystroglycan-sarcoglycan proteins at the sarcolemma[20].

**Loss of Thbs3 protects the heart from pressure overload.** Consistent with the greater cardiac disease profile observed in Thbs3 DTG mice, we next examined *Thbs3*$^{-/-}$ mice and observed partial protection from cardiac disease-inducing stimuli. For example, *Thbs3*$^{-/-}$ mice showed significantly less cardiac hypertrophy following 12 weeks of TAC stimulation compared with wildtype (WT) mice (Fig. 4a, b). *Thbs3*$^{-/-}$ mice also showed significantly less reduction in ventricular performance and less

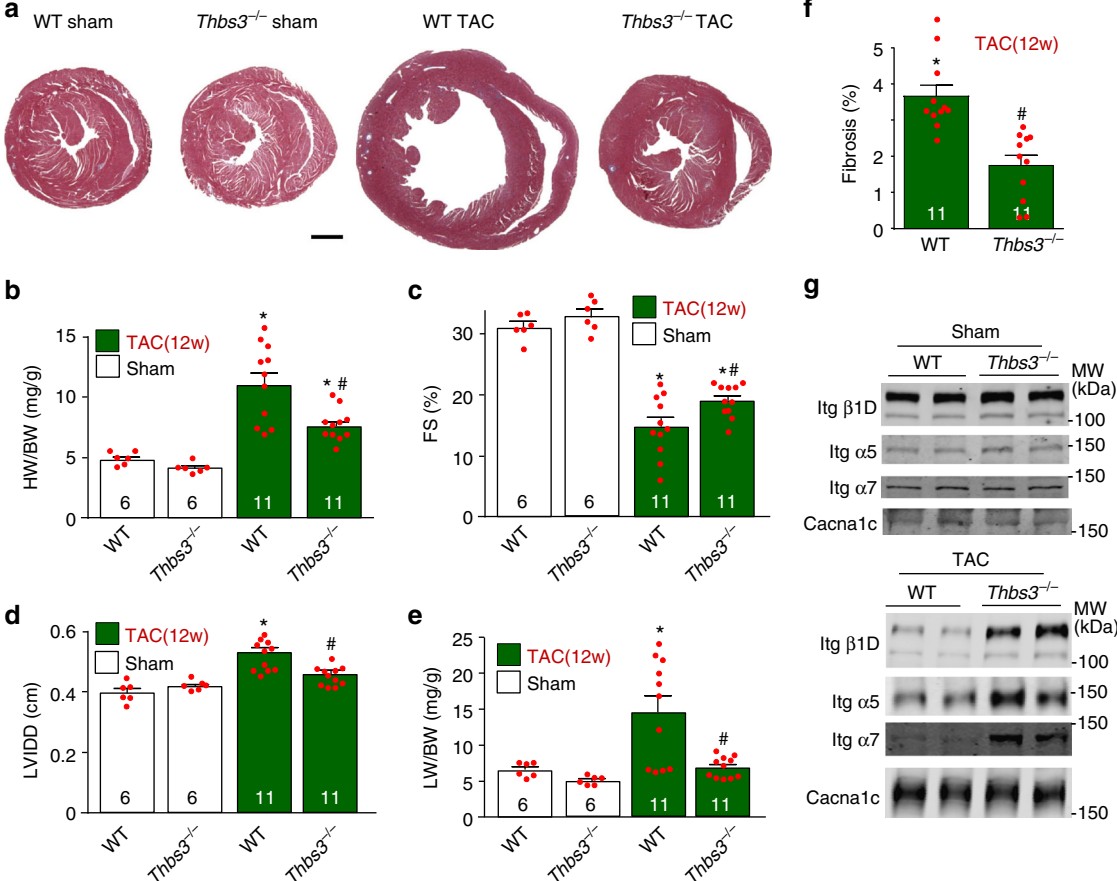

**Fig. 4** Loss of Thbs3 protects the heart from pressure overload pathology. **a** Low magnification cardiac histological images from WT and $Thbs3^{-/-}$ mice stained with Masson's trichrome 12 weeks after TAC or sham surgery. Scale bar = 1 mm. **b** HW/BW ratios in the indicated groups of mice 12 weeks after TAC or sham surgery. **c** Echocardiography measured fractional shortening (FS) percentage and **d** LVIDD in the indicated groups of mice 12 weeks after a TAC or sham surgical procedure. **e** Pulmonary edema was analyzed by LW/BW ratio measurement in the indicated groups of mice 12 weeks after a TAC or sham surgical procedure. **f** Percentage of fibrotic area was quantified with heart sections stained with Masson's trichrome in WT and $Thbs3^{-/-}$ mice after TAC. **g** Representative Western blots for integrin proteins from heart sarcolemma protein extracts from WT and $Thbs3^{-/-}$ mice 12 weeks after TAC or sham surgery. Cacna1c served as loading control. Quantitation of these results is shown in Supplementary Figure 4a-f. *$P < 0.05$ versus WT control sham; #$P < 0.05$ versus WT TAC. Statistical analysis was performed using one-way ANOVA and Turkey multiple comparisons test and two-tailed students $T$-test. Error bars are +/− standard error of the mean and number of mice used in each experiment are shown in the graphs

ventricular dilation over 12 weeks of TAC compared with WT control mice (Fig. 4c, d), as well as significantly less lung congestion and cardiac fibrosis (Fig. 4e, f). Mechanistically, while sham-operated mice showed no differences in integrin membrane protein levels in $Thbs3^{-/-}$ or WT mice, with TAC stimulation loss of Thbs3 resulted in greater levels of β1D, α5, and α7 integrin in the sarcolemma compared to WT controls with TAC (Fig. 4g and Supplementary Fig. 4a–f). These results provide evidence that endogenous Thbs3 upregulation in the heart during disease correlates with loss of integrin membrane residence.

**Induction of Thbs3 is detrimental after cardiac injury.** While deletion of *Thbs3* in the mouse was cardioprotective and led to greater sarcolemmal residence of select integrin subunits after TAC, we formulated another means of assessing the physiologic function of Thbs3 induction using combinatorial Thbs gene-deleted mice. Here we compared quadruple $Thbs1/2/4/5^{-/-}$ mice that can only upregulate Thbs3 during disease versus quintuple $Thbs1/2/3/4/5^{-/-}$ mice that are devoid of all Thbs genes (Fig. 5a). Other than an early neonatal growth delay, the quadruple and quintuple Thbs gene-deleted mice showed no cardiac phenotypic

maladies up to 1 year of age (Supplementary Fig. 5a–d). Western blotting from postnatal day 1–3-old neonatal mouse hearts confirmed the loss of each Thbs gene in the combinatorial deleted mice but also that $Thbs1/2/4/5^{-/-}$ mice showed a compensatory upregulation of endogenous Thbs3 (Fig. 5b).

We subjected WT, $Thbs1/2/3/4/5^{-/-}$ and $Thbs1/2/4/5^{-/-}$ mice to 2 weeks of TAC, which showed complete mortality in the $Thbs1/2/4/5^{-/-}$ mice by day 14, yet the WT and quintuple gene-targeted mice showed 90% and 77% survival over 14 days, respectively (Fig. 5c). No lethality was observed in sham operated mice (Fig. 5c). To also investigate cardiac sarcolemmal stability EBD uptake after Iso injection was measured in these same groups of mice (Fig. 5d). No significant differences were observed between WT and $Thbs3^{-/-}$ mice (Fig. 5e). However, $Thbs1/2/4/5^{-/-}$ showed significantly greater EBD positive cardiomyocyte area in the heart compared to WT, and further deletion of *Thbs3* in $Thbs1/2/3/4/5^{-/-}$ mice (quintuple deleted), were now largely rescued from this enhanced EBD uptake profile observed in $Thb1/2/4/5^{-/-}$ mice (Fig. 5e). Mechanistically, hearts from $Thbs3^{-/-}$ mice showed increased plasma membrane β1D integrin levels after TAC by immunohistochemistry compared to WT, while hearts from $Thbs1/2/4/5^{-/-}$ mice with only

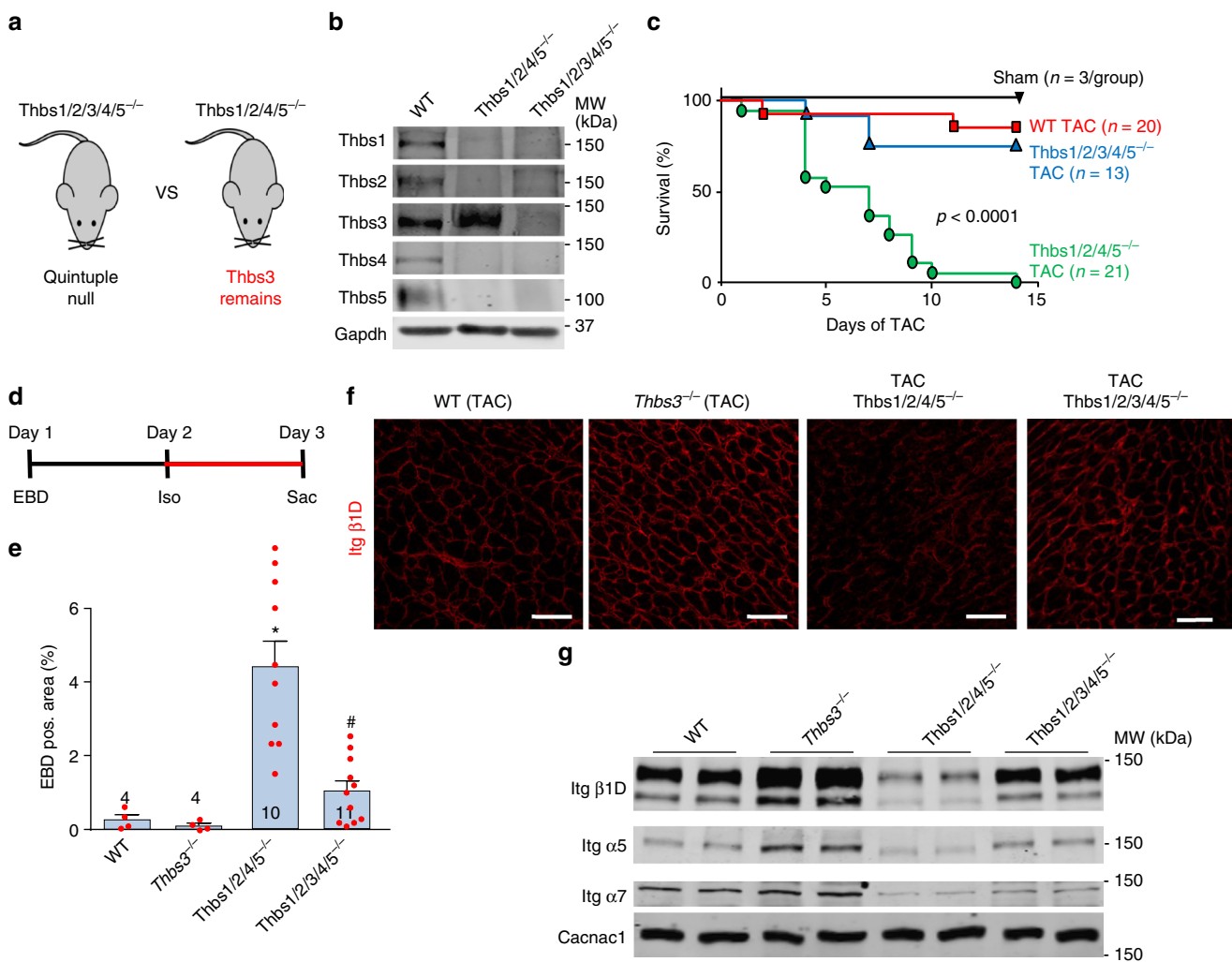

**Fig. 5** Induction of endogenous Thbs3 is detrimental after cardiac injury. **a** Schematic diagram depicting the Thbs gene-deleted mice used to selectively analyze effects of endogenous Thbs3. **b** Representative Western blots from hearts of 1–3 day-old WT, *Thbs1/2/3/4/5*−/− and *Thbs1/2/4/5*−/− mice for the indicated Thbs proteins. Gapdh served as loading control. **c** Kaplan–Meier survival plot of shams, WT, *Thbs1/2/3/4/5*−/− and *Thbs1/2/4/5*−/− mice after TAC surgery in days. Number of mice used is shown in the graph for each group. *P* < 0.0001 analyzed by log-rank test WT versus *Thbs1/2/4/5*−/− and *Thbs1/2/3/4/5*−/− versus *Thbs1/2/4/5*−/−. **d** Experimental regimen of EBD and Iso injection into the groups of mice shown (**e**) to measure membrane permeability. **e** Quantification of EBD positive area in the hearts of the indicated groups of mice after Iso injection with the regimen shown in **d**. *$P < 0.05$ versus WT; #$P < 0.05$ versus *Thbs1/2/4/5*−/− mice. Statistical analysis was performed using one-way ANOVA and Turkey multiple comparisons test. Error bars are +/− standard error of the mean and number of mice used in each experiment are shown in the graphs. **f** Immunohistochemistry for β1D integrin from heart sections of WT, *Thbs3*−/−, *Thbs1/2/4/5*−/− and *Thbs1/2/3/4/5*−/− mice 1 week after TAC surgery. Scale bars are 50 μm. **g** Representative Western blots for the indicated integrin proteins from hearts of WT, *Thbs3*−/−, *Thbs1/2/4/5*−/−, and *Thbs1/2/3/4/5*−/− mice subject to 1 week of TAC and processed for sarcolemma protein extracts. Cacna1c served as loading control. Quantitation of these results is shown in Supplementary Figure 4g–i

endogenous *Thbs3* showed reduced β1D integrin levels while *Thbs1/2/3/4/5* quintuple null mice were unaffected (Fig. 5f). Indeed, western blots for sarcolemmal α5, α7, and β1D integrin showed increased levels in *Thbs3*−/− hearts compared to WT controls after 1 week of TAC, yet hearts from *Thbs1/2/4/5*−/− mice showed inhibition of sarcolemmal residence of these same integrins and *Thbs1/2/3/4/5* quintuple null mice were unaffected (Fig. 5g and Supplementary Figure 4g–i). These data indicate that induction of endogenous *Thbs3* with cardiac injury reduces survival of mice after TAC and impairs sarcolemmal stability by likely reducing the cell surface residency of select integrins.

**Integrin overexpression reduces Thbs3-augmented disease**. To more directly prove that reduced integrin membrane content underlies cardiomyopathy associated with Thbs3 induction we

employed cardiac-specific transgenic mice overexpressing dimers of α7β1D integrin. The dual α7β1D transgenic approach was previously characterized and shown to have protective effects in cardiac and skeletal muscle disease[33,34]. Thbs3 DTG mice were crossed to mice with cardiomyocyte-specific overexpression of integrin α7 and β1D integrin (Fig. 6a), producing highly elevated levels of α7β1D even with Thbs3 overexpression (Fig. 6b). Interestingly, overexpression of α7β1D also lead to induction of endogenous integrin α5, inducing a fibronectin-binding integrin receptor (α5β1D), although total membrane levels of an unrelated membrane protein (α1Na$^+$/K$^+$ ATPase) was unaffected (Fig. 6b). Quantification of the EBD positive cardiac area after Iso injection again showed a significant increase in Thbs3 DTG hearts compared to tTA controls (Fig. 6c, d). However, overexpression of α7β1D in the hearts of Thbs3 DTG mice restored sarcolemmal

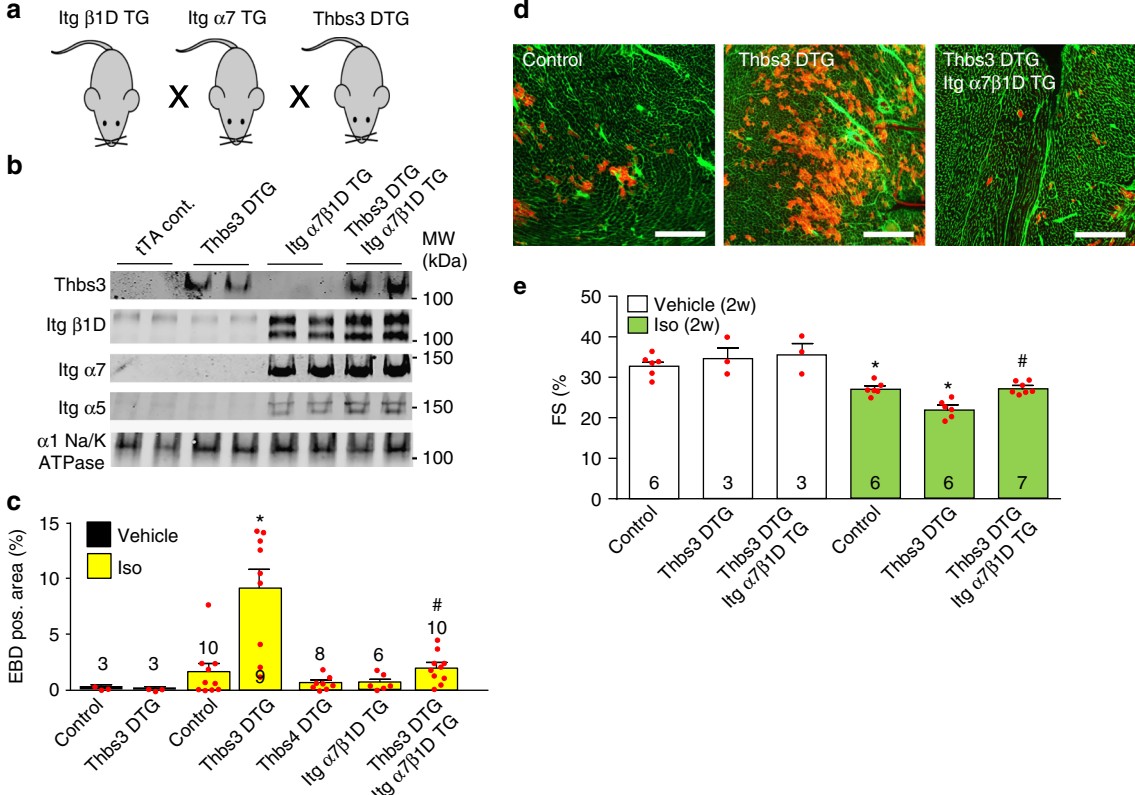

**Fig. 6** Integrin overexpression reduces Thbs3-mediated membrane instability and disease. **a** Schematic of the breeding used to generate combinatorial Thbs3 DTG, α7β1D integrin TG mice. **b** Representative Western blots for Thbs3 and the indicated integrin proteins from heart sarcolemma protein extracts from tTA, Thbs3 DTG, α7β1D integrin TG and Thbs3 DTG/ α7β1D integrin TG mice. The α1Na$^+$/K$^+$-ATPase served as loading control. **c** Quantification of EBD positive area from cardiac histological sections after Iso (300 mg/kg) injection in Thbs3 DTG, α7β1D integrin TG, Thbs4 DTG, and Thbs3 DTG/ α7β1D integrin TG mice. **d** Representative histological images of heart sections from tTA, Thbs3 DTG and Thbs3 DTG/ α7β1D integrin TG mice stained with WGA-FITC (green) after Iso injection (300 mg/kg). EBD is shown as red fluorescence. Scale bars are 300 μm. **e** Fractional shortening (FS) percentage as determined by echocardiography following 2 weeks of continuous Iso infusion (60 mg/kg/day) or PBS vehicle controls. Number of mice analyzed is shown within each histogram in **c**, **e**. *$P < 0.05$ versus vehicle treated; #<0.05 versus Thbs3 DTG with Iso. Statistical analysis was performed using one-way ANOVA and Turkey multiple comparisons test. Error bars are $+/-$ standard error of the mean

stability and inhibited the increase in EBD positive area induced by Thbs3 overexpression (Fig. 6c, d). No differences in EBD uptake were observed between control and Thbs4 DTG animals after Iso injection (Fig. 6c). We also used a model whereby Iso was infused for 2 weeks to induce cumulative disease, which caused a significantly greater loss in cardiac ventricular performance in Thbs3 DTG mice that was prevented by the presence of the α7β1D transgenes (Fig. 6e). Taken together, these results demonstrate that overexpression of integrin α7β1D is sufficient to rescue Thbs3-mediated membrane instability and greater disease manifestation with pathological stimulation, suggesting a primary mechanism whereby Thbs3 serves its maladaptive role by directly inhibiting integrin sarcolemmal residence on the cardiomyocyte.

**Thbs3 blocks integrin membrane trafficking by an EGD motif.**
We showed previously that Thbs4 overexpression augments intracellular protein trafficking, leading to greater membrane residence of attachment complexes but also greater levels of select ECM proteins in skeletal muscle, presumably due to ATF6α induction and chaperone activity of Thbs4[21]. Similarly, here we observed that overexpression of Thbs3 or Thbs4 using adenoviral gene transfer similarly enhanced global protein trafficking from ER-to-Golgi and exit from the Golgi to the sarcolemma compared with Adβgal infected NRVMs (Fig. 7a, b). These results are consistent with the observed expansion of ER and post-ER vesicles

documented by transmission electron microscopy (Fig. 1h and Supplementary Fig 1d), and with an observed increase in the ECM content of lumican, fibronectin, fibromodulin and collagen 4 with either Thbs3 or Thbs4 transgene-mediated overexpression in the heart at baseline or with TAC (Fig. 7c). While general ER-to-Golgi protein trafficking (assayed in NRVMs with a baculovirus expressing GalNacT2-RFP) and Golgi-to-sarcolemma protein trafficking (assayed with a temperature activated VSVG-GFP mutant protein) was similarly augmented with Thbs3 and Thbs4, differential trafficking was observed with a fluorescently tagged α5 integrin protein in live cells. For example, Thbs3 significantly reduced α5 integrin exiting the Golgi whereas Thbs4 significantly enhanced it (Fig. 7d). Importantly, addition of extracellular, recombinant Thbs proteins did not alter α5 integrin trafficking between the groups indicating that integrin membrane trafficking was regulated intracellularly (Fig. 7e). Importantly, the same recombinant Thbs3 and Thbs4 protein preparations used here promoted luciferase refolding to a level comparable to the canonical chaperone heat shock protein 70 (Hsp70) (Supplementary Fig. 6a, b). The chaperone activity was specific to Thbs3/4 proteins as Nell2, a closely related secreted pentameric glycoprotein with some conserved domains, lacked chaperone refolding activity (Supplementary Fig. 6a, b)[24,35]. These results indicate that Thbs proteins have direct and unique chaperone activity but they have no effect on integrin membrane trafficking when added to the outside of cells.

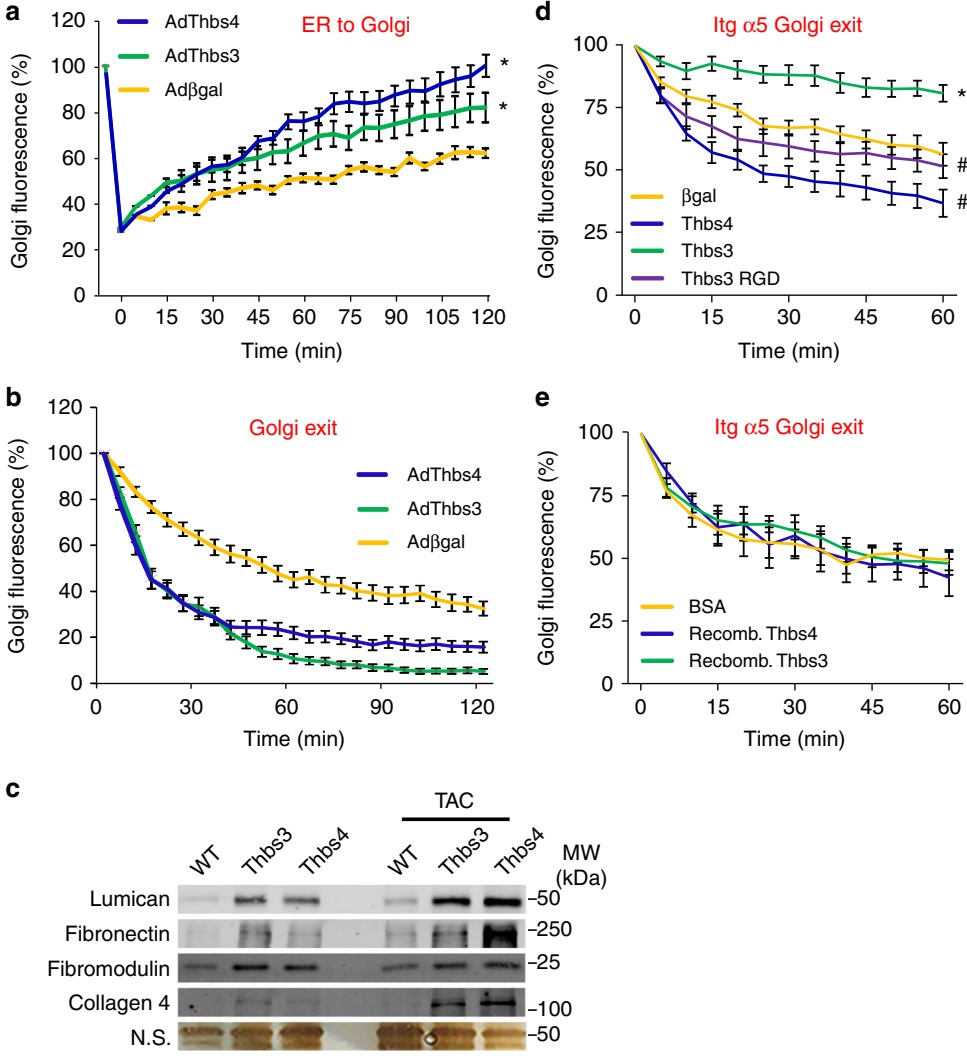

**Fig. 7** Thbs3 enhances secretory pathway activity but reduces membrane integrins. **a** Time course of intracellular trafficking fluorescence changes in cultured neonatal ventricular cardiomyocytes (NRVMs) infected with a GalNac-T2-RFP baculovirus and the indicated adenoviruses. The data show a quantitative time course of GalNac-T2-RFP recovery in the Golgi network after FRAP to measure ER-to-Golgi vesicular trafficking. *$P < 0.05$ versus Adβgal infected cells. **b** Quantitative time course of loss of VSVG-eGFP fluorescence in the Golgi after iFRAP as a measurement for Golgi-to-membrane (Golgi exit) vesicular trafficking. *$P < 0.05$ versus Adβgal infected cells. Statistical analysis was performed using two-way ANOVA and Turkey multiple comparisons test. **c** Western blots for ECM proteins using heart extracellular matrix extracts from tTA, Thbs3 DTG and Thbs4 DTG mice at baseline or with TAC stimulation. A silver-stained gel is shown with a non-specific (n.s.) band as a loading control. **d**, **e** Quantitative time-course of the loss of α5 integrin-GFP fluorescence at the Golgi after iFRAP as a measure of Golgi-to-plasma membrane vesicular trafficking of α5 integrin. The data were generated by iFRAP of α5 integrin-GFP from COS-7 cells **d** transfected with the plasmids harboring the cDNAs shown or **e** treated with recombinant Thbs3 or Thbs4 proteins, or bovine serum albumin (BSA) as a control. *$P < 0.05$ versus Adβgal transfected cells; #$<0.05$ versus Thbs3 transfected cells. Statistical analysis was performed using two-way ANOVA and Turkey multiple comparisons test. Results were summed from four independent experiments and error bars are +/− standard error of the mean

Comparison of the highly conserved C-terminal domains of Thbs3 with other Thbs proteins revealed distinct differences in integrin binding motifs contained in the type-3 repeats (Supplementary Fig. 7a). All Thbs proteins except Thbs3 contain conserved RGD or KGD integrin binding motifs in this region[30]. However, Thbs3 contains a conserved EGD sequence at the same relative position as the canonical integrin binding RGD/KGD sites found in Thbs4/5 (Supplementary Fig. 7a). Mutation of the EGD motif in Thbs3 to the RGD sequence (Thbs3 RGD) did not alter its secretion (Supplementary Fig. 7b) or localization (Supplementary Fig. 7c) when overexpressed in NRVMs. Remarkably, analysis of this Thbs3 RGD mutant construct revealed that it abrogated the inhibitory function of Thbs3 on α5 integrin trafficking and significantly enhanced α5 integrin

exiting from the Golgi complex compared to WT Thbs3 transfected cells (Fig. 7d).

To further characterize the differences between Thbs3, Thbs4 and Thbs3 RGD, we performed cell surface biotinylation experiments using NRVMs overexpressing these Thbs proteins. As expected, Thbs3 reduced cell surface levels of α5 and β1 integrin compared to βgal or Thbs4, and more importantly the Thbs3 RGD mutant now restored levels of these integrins at the cell surface in infected NRVMs (Fig. 8a–d). Moreover, ER resident chaperones CRT and CNX were found at increased levels at the cell surface in Thbs3 overexpressing NRVMs but not with Thbs4 or Thbs3 RGD overexpression (Fig. 8a–d). CRT and CNX are critical chaperones that normally facilitate integrin membrane placement suggesting that their unique presence at the

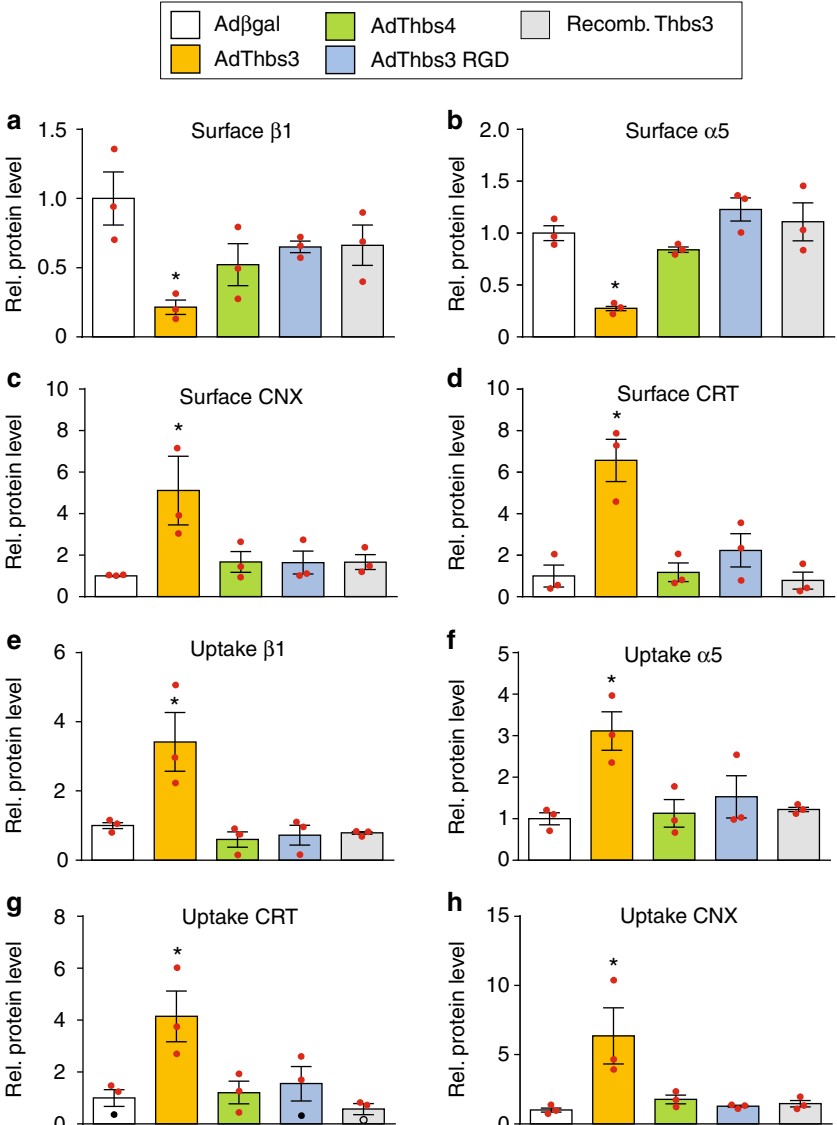

**Fig. 8** Thbs3 enhances integrin turnover. **a–h** Cell surface biotinylated proteins were used for input control experiments or immunoprecipitated with streptavidin (Surface, **a–d**) and endocytic protein uptake was induced for 30 min followed by immunoprecipitation with streptavidin (Uptake, **e–h**). Quantitative analysis of protein band intensities of β1D integrin, α5 integrin, calreticulin (CRT), and calnexin (CXN) from NRVMs infected with Adβgal, AdThbs3, AdThbs4, AdThbs3-RGD adenovirus or treated with recombinant Thbs3 protein (western blots used for quantification are presented in the uncropped western blot images attachment). Data are represented as fold expression over Adβgal relative to input controls. *$P < 0.05$ versus Adβgal. Statistical analysis was performed using one-way ANOVA and Turkey multiple comparisons test. Error bars are $+/-$ standard error of the mean and number of independent experiments are shown in the graphs

sarcolemma only with Thbs3 was facilitating reorganization of the integrin complexes[36–38]. Indeed, analysis of endocytic protein uptake after cell surface biotinylation revealed that Thbs3 enhanced integrin uptake and also endocytosis of CRT and CNX, which was not observed with Thbs4 or Thbs3 RGD (Fig. 8e–h), suggesting that Thbs3 also promotes greater turnover of sarcolemmal integrin content.

## Discussion

Thbs proteins have been ascribed beneficial effects in the heart after injury. For example, $Thbs1^{-/-}$ mice showed severe phenotypic defects after cardiac injury[39–41] and $Thbs2^{-/-}$ mice are more prone to cardiac ventricular wall rupture with pathologic stimulation[16,42,43], while $Thbs4^{-/-}$ mice show reduced collagen maturation or decompensation with injury[17–19]. These enhanced disease profiles in $Thbs1^{-/-}$, $Thbs2^{-/-}$, and $Thbs4^{-/-}$ mice have

been associated with defects in the ECM, although it remains unclear whether this is due to extracellular or intracellular functionality of these proteins. For example, we have shown that Thbs1 and Thbs4 have important functions inside the cell by regulating an ATF6α-mediated ER-stress response, vesicular expansion and enhanced protein transport within the secretory pathway[19,21,24]. This increased protein trafficking results in protection from cardiac and skeletal muscle injury[19,21]. Moreover, Thbs1 has been shown to protect pancreatic β-cells via an intracellular PKR-like ER kinase-nuclear factor E2 related factor 2 (PERK-NRF2) pathway[44].

Even though Thbs3 is expressed in the heart[14,15] and is upregulated after cardiac injury[9,16], its role in this organ was largely unknown. We observed that Thbs3 activates ATF6α with augmentation in ER resident stress proteins and ER and post-ER vesicular expansion, similar to previous observations with Thbs4

in heart and skeletal muscle[19–21]. However, Thbs3 is unique in that it functions opposite to Thbs1, Thbs2, and Thbs4, where it enhances cardiac pathology with disease stimulation, while mice deficient in $Thbs3^{-/-}$ were partially protected from cardiac disease.

Mechanistically, overexpression of Thbs3 in cardiomyocytes of the mouse heart dramatically reduced sarcolemmal residence of nearly all integrin subunits assayed, which destabilized the membrane and lead to greater cardiac pathology with stress/disease stimulation. Indeed, overexpression of the integrin α7β1D heterodimers is protective in heart and skeletal muscle diseases[33,34,45], whereas deletion of integrin β1 in the heart induces cardiomyopathy[32]. Finally, the greater cardiac disease profile associated with Thbs3 overexpression was corrected by α7β1D heterodimer overexpression, directly demonstrating a causal relationship.

The effect of Thbs3 toward inhibiting integrin sarcolemmal residence is likely not due to an artifact of overexpression since Thbs4 overexpression had no such effect, and in fact, Thbs4 enhanced sarcolemmal residence of select integrin subunits and β-dystroglycan. Moreover, $Thbs1/2/4/5^{-/-}$ mice that only express endogenous Thbs3 with injury showed the same membrane stability defect with greater pathology and premature lethality following cardiac disease stimuli, which was also associated with reduced integrins at the sarcolemma because $Thbs1/2/3/4/5^{-/-}$ mice showed a near complete rescue of these effects. However, it should be noted that mice only containing the $Thbs3$ gene ($Thbs1/2/4/5^{-/-}$ mice) is an artificial situation, as during cardiac injury all 5 Thbs genes are induced and likely contribute to the final healing process.

The results obtained here suggest a very dynamic role for each of the Thbs family members in disease and tissue healing, with Thbs3 clearly playing an antithetical role compared with Thbs1/2/4/5. Thbs3 enhances secretion of certain ECM proteins while at the same time it reduces sarcolemmal integrin levels. The combination of enhanced ECM secretion and reduced integrin binding of the embedded cells might assist in aspects of cellular remodeling with tissue injury or wound healing as cells reposition themselves in the scar or hypertrophic ventricle. Indeed, integrin shedding was proposed as a mechanism to allow transient remodeling of the cardiomyocyte-ECM connection during hypertrophic remodeling[46]. Thus, upregulation of Thbs3 would allow cardiomyocytes to remodel their interaction with the extracellular environment for a period of time, which likely becomes pathologic if chronically employed due to continued sarcolemmal instability associated with the de-attachment process (Supplementary Fig. 8).

The direct interaction between Thbs proteins and a wide variety of integrins has been well established in the literature, perhaps more so than for any other Thbs interacting protein[28–30]. Thbs1 harbors de-adhesive properties by interacting with CRT and LRP1 at the cell surface[47], permitting an intermediate state of adhesion[48,49]. However, our data are most consistent with a model whereby Thbs3 reduces cellular adhesion and membrane stability from inside the cell by reducing integrin trafficking and by enhancing protein turnover by increasing CRT and CNX levels at the cell surface[50,51]. Application of exogenous recombinant Thbs3 or Thbs4 had no effect on integrin membrane trafficking or residence, yet recombinant Thbs4 and Thbs3 showed strong protein chaperone activity in vitro. Indeed, Thbs proteins may conduct many of their known biologic effects from within the vesicular trafficking network through their chaperone activity, including controlling ECM composition and membrane surface protein placement. Such an interpretation is also consistent with carefully conducted in vivo imaging studies that consistently fail to show Thbs proteins within the ECM[19,52–58]. In our studies,

which verified antibody specificity in gene-deleted mice or with transgene-mediated overexpression, Thbs1, Thbs3, and Thbs4 are observed within the vesicular compartment of cells in heart and skeletal muscle, but not within the ECM region[19,21].

The canonical integrin recognition sequence is RGD[59], which is found in most Thbs proteins across animal species. Human and mouse Thbs4 contain a RGD domain in the third type-3 repeat[60], while *Drosophila melanogaster* Tsp and *Danio rerio* Thbs4 have integrin binding activity via their KGD and RGD domains that are in similar relative positioning within the type 3 repeats[61,62]. However, Thbs3 contains a conserved EGD domain across most vertebrate species that generally aligns with the RGD domain in Thbs4 (Supplementary Fig. 5a). Mutating the Thbs3 EGD domain to RGD now reverses the negative regulatory effect of Thbs3 on integrins, allowing proper membrane trafficking with reduced turnover through CRT and CNX. We also recently mutated Thbs4 so that it was not secreted or processed through the sarcolemma, which specifically inhibited membrane residence of β-dystroglycan, β-sarcoglycan, and δ-sarcoglycan from the dystrophin-glycoprotein attachment complex, promoting membrane instability and cardiomyopathy in mice[20]. Since Thbs4 appears to preferentially chaperone the dystroglycans and sarcoglycans to the sarcolemma[21], it is consistent with the general paradigm proposed for Thbs3 as a regulator of integrin membrane trafficking, further suggesting that the entire Thbs gene family can affect cell-membrane attachment versus release dynamics during tissue injury and healing (Supplementary Fig. 8).

## Methods

**Transgenic animals**. All experimental procedures with animals were approved by the Institutional Animal Care and Use Committee of Cincinnati Children's Medical Center, protocols IACUC 2015-0047 and 2016-0069. We have complied with the relevant ethical considerations for animal usage overseen by this committee. The number of mice used in this study reflects the minimum number needed to achieve statistical significance based on experience and previous power analysis. Blinding was performed for some experimental procedures with mice, although blinding was not possible in every instance. Both sexes of mice were used.

Cardiomyocyte-specific transgenic mice were generated using the bigenic tetracycline murine α-myosin heavy chain (MHC) promoter expression vector system[27]. Full-length mouse Thbs3 cDNA was obtained from Harvard Plasmids (Plasmid ID: MmCD00319977), amplified by PCR and cloned in the HindIII site of the α-MHC responder vector (forward 5′-aaaaaaagcttatggagaagccggaacttt-3′, reverse 5′-aaaaaaagctttcacactcttccctggagc-3′). The final construct was confirmed by DNA sequencing and injected into newly-fertilized oocytes to generate transgenic animals at the University of Cincinnati Transgenic Mouse Core (C57Bl/6J background). Mice harboring the α-MHC-Thbs3 transgene were crossed to mice expressing the tetracycline trans-activator (tTA) under the control of the α-MHC promoter to generate double transgenic animals (DTG) and maintained on a C57Bl/6J background[27]. Mice harboring the α-MHC-Thbs4 transgene were backcrossed to a C57Bl/6J background 6 times[19]. Thbs3 overexpressing animals were crossed to integrin α7/β1D overexpressing transgenic animals[34] and genotyped with gene specific primers. The Thbs3 β-galactosidase reporter mouse strain was created from ES cell clone 19211A-A12, generated by Regeneron Pharmaceuticals, Inc. and obtained from the KOMP Repository (www.komp.org). Embryonic stem cell aggregation with 8-cell embryos was used to generate chimeric mice. Germline transmitting male chimeras were crossed with C57Bl/6J females to establish the colony. Mice deficient for Thbs4 ($Thbs4^{-/-}$; Strain: B6.129P2-Thbs4tm1Dgen/J) and Thbs1 ($Thbs1^{-/-}$; Strain: B6.129S2-Thbs1tm1Hyn/J) were purchased from Jackson Laboratories. Thbs3 and Thbs5 gene-deleted mice were provided by Dr. J.T. Hecht[12,13]. Thbs2 deficient mice were provided by Dr. T. Kyriakides[63]. Mice were backcrossed to a C57Bl/6J background 6 times and genotyped with gene specific primers. Thbs deficient animals were intercrossed to generate quintuple $Thbs1/2/3/4/5^{-/-}$ and quadruple $Thbs1/2/4/5^{-/-}$ gene-targeted animals. No human subjects or human tissue was used in this study.

**Neonatal rat cardiomyocytes**. Primary neonatal rat ventricular cardiomyocytes (NRVMs) were isolated from 1-day to 2-day old Sprague-Dawley rat pups (Envigo, #002 timed mated), cultured on gelatin-coated dishes[19,64], and maintained in M199 medium (ThermoFisher Scientific, #SH30253FS) supplemented with 2% bovine growth serum (ThermoFisher Scientific, #SH3054103) and 1× penicillin-streptomycin (Cellgro, #30-0002-CI) at 37 °C in 5% $CO_2$. For the isolation procedure, neonatal hearts were collected, the atria were removed, and the ventricles

were cut up in HBSS prior to enzymatic digestion. The ventricular tissue was subjected to 5 rounds of enzymatic digestion using 0.05% pancreatin (Sigma) and 84 units/ml of collagenase (Worthington, Lakewood, NJ). Cells were collected by centrifugation at 500×g for 5 min at 4 °C and resuspended in M199 medium. The cells were then differentially plated for 1 h on culture dishes to reduce fibroblasts. The cardiomyocytes were then plated on gelatinized cell culture dishes.

**COS-7 cells**. COS-7 cells (ATCC, #CRL-1651) were maintained in DMEM/high glucose (ThermoFisher Scientific, #SH30022.01) supplemented with 10% bovine growth serum (ThermoFisher Scientific, #SH3054103) and 1× penicillin-streptomycin (Cellgro #30-0002-CI) at 37 °C in 5% $CO_2$.

**Adenoviruses**. The Thbs3 cDNA was sub-cloned into the pShuttle-CMV (Agilent Technologies #240007) vector to produce recombinant adenovirus following manufacturer's instructions. The pShuttle-CMV-Thbs3 construct was used to generate the Thbs3 RGD mutant (5′-cacaggcatctcctcttccattgccatccgtgtcct-3′) using the QuikChange II Site-Directed Mutagenesis Kit (Agilent Technologies, #200522). Adenoviruses harboring Thbs4, the endoplasmic reticulum (ER) luminal domain of activating transcription factor 6α (ATF6α, amino acids 448–570) with a C-terminal ER KDEL retention signal (Ad-ATF6α-DN-Myc-KDEL) and β-galactosidase (βgal) expressing control were previously generated and validated[24]. The eGFP-tagged VSVG-ts045 (VSVG-eGFP, Addgene: Plasmid #11912) cDNA was amplified by PCR for insertion into the pAdenoX-CMV vector (Clontech, #632269) and transfected into HEK cells to generate recombinant adenovirus following manufacturer's instructions[24]. NRVMs were grown in Medium 199/EBSS (ThermoFisher Scientific, #SH30253FS) supplemented with 2% bovine growth serum (Thermo-Fisher Scientific, # SH3054103) and 1% penicillin-streptomycin (Cellgro, #30-0002-CI). NRVMs were then infected with adenovirus for 2 h and then fresh media was applied. Cells were harvested 24 to 72 h post-infection.

**Protein isolation and Western blotting**. Hearts were excised, frozen in liquid nitrogen and stored at −80 °C. Whole ventricle proteins samples were prepared in a modified radio-immunoprecipitation assay (RIPA) buffer (1% Triton X-100, 1% sodium deoxycholate, 0.1% SDS, 50 mM Tris-HCl, pH 7.4, 150 mM NaCl and protease inhibitors cOmplete Mini, EDTA-free Protease Inhibitor Cocktail (Millipore Sigma, #11836170001)). Homogenates were centrifuged at 14,160×g for 10 min at 4 °C and supernatants were stored at −80 °C until further use.

Crude sarcolemma protein isolates were prepared from mouse heart[65]. Briefly, hearts were homogenized in ice-cold lysis buffer (20 mM $Na_4P_2O_7$, 20 mM $NaH_2PO_4$, 1 mM $MgCl_2$, 0.3 M sucrose, 0.5 mM EDTA, pH 7.1 with protease inhibitors), then centrifuged at 16,870×g for 10 min at 4 °C, supernatants were then centrifuged at 30,000×g for 30 min at 4 °C after which the pellet was re-suspended in 100 µl RIPA buffer and stored at −80 °C until further use. Integrin-β1D positive intracellular vesicles were isolated from hearts using an ER isolation kit (Millipore Sigma, #ER0100), according to the manufacturer's instructions. Briefly, NRVMs were homogenized in isotonic extraction buffer using a 2 ml Dounce homogenizer. Homogenates were cleared by centrifugation at 16,870×g for 15 min at 4 °C. Two milligrams of crude microsomal fraction were incubated with an antibody raised against the cytoplasmic domain of β1D-integrin (EMD Millipore, #MAB1900) for 12 h at 4 °C, immunoprecipitated using protein A/G magnetic beads (ThermoFisher Scientific, #88803) at 4 °C for 1 h, and then subjected to SDS-PAGE.

The protocol for immunoprecipation (IP) was from the Abcam website (http://www.abcam.com/protocols/immunoprecipitation-protocol-1). NRVMs were lysed in non-denaturing lysis buffer (1% Triton X-100, 50 mM Tris-HCl, pH 7.4, 150 mM NaCl containing protease inhibitors). The IP was performed with an anti-c-Myc antibody (Santa Cruz Biotechnology, #sc40 at 2 µg antibody/mg protein) and Protein A/G magnetic beads (ThermoFisher Scientific, #88803).

Extracellular protein fractionation from heart tissue was performed as previously described[66]. Briefly, hearts were minced and washed in PBS with 25 mM EDTA. Tissue was decellularized overnight in 0.1% SDS, 25 mM EDTA, followed by protein solubilization in 4 M guanidine hydrochloride, 50 mM $C_2H_3NaO_2$, 25 mM EDTA (pH 5.8). Proteins were precipitated overnight in 80% EtOH, air dried and treated with protein deglycosylation mix (New England Biolabs, #P6039).

Cell surface biotinylation and analysis of endocytic uptake was performed[67]. In brief, NRVMs were infected with Adβgal control, AdThbs3, AdThbs3 RGD or AdThbs4 adenoviruses. Cells were grown in Medium 199/EBSS (ThermoFisher Scientific, #SH30253FS) supplemented with 1% penicillin-streptomycin (Cellgro, #30-0002-CI) for 48 h. Recombinant Thbs3 (R&D Systems, #8390-TH-050) was added at a final concentration of 1 µg/ml, 24 h before the start of the experiment. After three washing steps in ice cold PBS, cells were incubated with 0.5 µg Sufo-NHS-SS-biotin (ThermoFisher Scientific, #21331) in PBS at 4 °C for 30 min. For the endocytic uptake assay, biotinylated cells were washed in warm (37 °C) PBS and incubated with warm (37 °C) Medium 199/EBSS supplemented with 100 mM primaquine bisphosphate for 30 min at 37 °C. The non-internalized biotin fraction at the cell surface was removed with 60 mM sodium 2-mercaptoethanesulfonate (MesNa (Sigma, M1511)) in MesNa buffer (50 mM Tris–HCl, pH 8.6, 100 mM NaCl) for 30 min at 4 °C, followed by quenching with 100 mM iodoacetamide (Sigma, #I6125) for 15 min at 4 °C. Surface labeled cells (surface) and cells that

were allowed to internalize proteins (uptake) were lysed in RIPA buffer (1% Triton X-100, 1% sodium deoxycholate, 0.1% SDS, 50 mM Tris-HCl, pH 7.4, 150 mM NaCl and protease inhibitors cOmplete Mini, EDTA-free Protease Inhibitor Cocktail (Millipore Sigma, #11836170001)). Protein lysates for input control were mixed with 5× Laemmli loading buffer and heated at 95 °C for 10 min. Equal amounts of protein (10 mg) from surface and uptake experiments were mixed with 25 µl Dynabeads M-280 streptavidin (ThermoFisher Scientific, #11205D) and incubated overnight at 4 °C. Biotinylated proteins were mixed with 1× Laemmli loading buffer, heated at 95 °C for 10 min and subjected to SDS-PAGE.

Secretion of Thbs proteins was analysed using NRVM. NRVMs were infected with Adβgal control, AdThbs3, AdThbs3 RGD or AdThbs4 adenoviruses. Cells were grown in Medium 199/EBSS (ThermoFisher Scientific, #SH30253FS) supplemented with 2% bovine growth serum (ThermoFisher Scientific, # SH3054103) and 1% penicillin-streptomycin (Cellgro, #30-0002-CI) for 48 h. Then, cells were washed in PBS and serum free Medium 199/EBSS was added. Media was collected after 4 h and concentrated using Amicon Ultra-15 centrifugal filter with Ultracel-30 membrane (EMD Millipore, #UFC903024). Concentrated media were mixed with 5× Laemmli loading buffer and equal amounts of protein were subjected to SDS-PAGE.

Protein concentrations were determined using a Direct Detect infrared spectrometer (Millipore Sigma). Protein preparations were mixed with 5× Laemmli loading buffer and heated at 95 °C for 10 min. Equal amounts of protein were subjected to SDS-PAGE and transferred to polyvinylidene fluoride (PVDF) membranes (EMD Millipore, #IPFL00010). Immunoblots were performed using appropriate primary antibodies and fluorescent conjugated secondary antibodies (LI-COR, IRdye 800CW Goat anti-Mouse #926-32350, IRdye 800CW Goat anti-Rat #926-32219, IRdye 800CW Goat anti-Rabbit #926-32211, IRdye 680RD Goat anti-Mouse #926-68072, IRdye 680RD Goat anti-Rabbit #925-68073 all at 1:3000) in combination with an Odyssey CLx Infrared Imaging System (LI-COR). Western blot band intensities were quantified using the Li-Cor Image Studio software.

Primary antibodies used were: glyceraldehyde 3-phosphate dehydrogenase (Gapdh (Fitzgerald, #10R-G109A at 1:10000)), Armet (Abcam, #ab67271 at 1:1000), ATF6α (Abcam, #ab37149 at 1:1000), calreticulin (Cell Signaling Technology, #2891; at 1:1000), calnexin (Abcam, #ab75801 at 1:1000), GRP78/BiP (Sigma-Aldrich, #G8918 at 1:1000), PDI (Cell Signaling Technology, #2446; at 1:1000), integrin β1D (EMD Millipore, #MAB1900 at 1:1000), integrin β3 (Cell Signaling Technology, #4702; at 1:1000), integrin α2 (Lifespan Biosciences, #LS-C159934 at 1:1000), integrin α4 (EMD Millipore, #AB1924 at 1:1000), integrin α5 (EMD Millipore, #AB1928 at 1:1000), integrin α6 (Cell Signaling Technology, #3750 at 1:1000), integrin α7 (Santa Cruz Biotechnology, #sc-27706 at 1:100), integrin α9 (Abcam, #ab140599 at 1:1000), integrin α10 (EMD Millipore, #AB6030 at 1:1000), Thbs1 (Lifespan Biosciences, #LS-B4155 at 1:1000), Thbs2 (BD Bioscience, #611150 at 1:1000), Thbs3 (Proteintech, #19727-1-AP at 1:1000), Thbs4 (Santa Cruz Biotechnology, #sc-7657 at 1:500), COMP/Thbs5 (Proteintech, #13641-1-AP at 1:1000), laminin2α2 (Sigma, #L0063 at 1:1000), β-tubulin (Licor, #926-42211 at 1:1000), Cacna1 (Alomone Labs, #ACC-022 at 1:1000), sodium-potassium ATPase (Abcam, #ab76020 at 1:1000).

All of the uncropped western blots contained in this report are shown in the supplementary information file as Supplementary Figure 9 which also shows sizes of molecular weight migration standards.

**Cardiac injury models and echocardiography**. Eight to eleven week-old mice of the relevant genotypes were subjected to transverse aortic constriction (TAC) or sham surgical procedure[68]. For this procedure, the transverse aortic arch was visualized through a median sternotomy and 7-0 silk ligature was tied around the aorta and a 27-gauge wire to control the degree of constriction between the right brachiocephalic and left common carotid arteries, after which the wire was removed to generate a defined reduction in lumen area. Doppler echocardiography was performed on mice after TAC to ensure equal pressure gradients across the aortic constriction. Mice with pressure gradients of less than 45 mmHg and fractional shortening greater than 30% at 2 weeks post-TAC were excluded from the results, as this indicated an unsuccessful surgery. Myocardial infarction (MI) was induced via permanent surgical ligation of the left coronary artery with 8-0 prolene suture, which was visualized through a medium sternotomy[69]. For both the TAC and MI surgical procedures mice were anesthetized with inhaled 2% isoflurane to effect (mice were intubated and respirated throughout). Both sexes of mice were used for TAC and MI surgeries. Mice were given buprenex (Henry Schein, #055175) as pain relief at a final concentration of 0.05 mg/ml by subcutaneous injection. Mice were then transferred to 30 °C incubators and monitored while in recovery, and any mouse displaying signs of distress was removed from the study in accordance institutional guidelines approved by the IACUC of Cincinnati Children's Hospital. Echocardiography was performed in a blinded manner on mice after isoflurane inhalation (2% to effect) for anesthesia using a Sonos 5500 ultra-sound instrument (Hewlett Packard) with a 15 MHz microprobe and measurements were taken in M-mode in triplicate for each mouse and averaged. Infusion of isoproterenol (60 mg/kg/day (Sigma Aldrich, #16504)) was performed with implantation of Alzet minipumps for 2 weeks (Durect, #7147160-6). For Evan's blue dye (EBD) experiments a single intraperitoneal (i.p.) isoproterenol injection of 300 mg/kg was given (see below).

**Vesicular trafficking and integrin-α5 trafficking**. Live imaging to evaluate integrin α5 trafficking was performed in COS-7 cells (ATCC, #CRL-1651). COS-7 cells were plated in 35 mm glass-bottomed gelatin-coated MatTek culture dishes (MatTek, #P35G-0-10-C) and maintained in DMEM/high glucose (ThermoFisher Scientific, #SH30022.01) supplemented with 10% bovine growth serum (Thermo-Fisher Scientific, #SH3054103) and 1× penicillin-streptomycin (Cellgro, #30-0002-CI) at 37 °C in 5% CO$_2$. When 50% confluent, COS-7 cells were transfected using Lipofectamine 3000 transfection reagent (Invitrogen, #L3000008) according to manufacturer instructions. More specifically, integrin α5-GFP (Addgene, #15238) was co-transfected either with pCMV-SPORT-βgal control (Invitrogen, #10586014), pShuttle-CMV-Thbs3-WT, pShuttle-CMV-Thbs3-RGD mutant or pCMV-Tag1-Thbs4[19]. Twenty-four hours after transfection, live cell quantitative imaging and photo bleaching was performed to evaluate integrin α5-GFP movement and vesicular trafficking (Golgi exit; iFRAP) using a Nikon A1+ confocal laser microscope system (Nikon Instruments) equipped with Plan Apo 40× oil immersion objective (Nikon Instruments, NA = 1.0), an INU-TIZ-F1 stage top incubator (Tokai hit) and NIS Elements AR microscope imaging software (Nikon Instruments) as previously described[21]. Briefly, 2 h prior to imaging, 200 μg/ml cycloheximide (Sigma Millipore, #C104450) was added to the cells. Bovine serum albumin (BSA, ThermoFisher Scientific, #BP1600), recombinant Thbs3 (R&D Systems, #8390-TH-050) and recombinant Thbs4 (R&D Systems, #7860-TH-050) were added at a final concentration of 1 μg/ml. The cargo pool in the Golgi was selectively highlighted by photo bleaching integrin α5-GFP from the entire cell excluding the perinuclear Golgi network using a high intensity laser at 488 nm (100% laser power). Time-lapse imaging (5% laser power) at 1-min intervals was then conducted for 60 min to monitor export of integrin α5-GFP molecules as a measurement of Golgi to membrane protein trafficking.

For general ER-to-Golgi and Golgi-to-membrane vesicular trafficking experiments, primary neonatal rat ventricular myocytes (NRVMs) were infected with Adβgal control, AdThbs3 or AdThbs4 adenoviruses in 6 cm dishes in 1 ml of media for 4 h, after which the dishes were washed 1× in fresh media and then incubated for 1–3 days in fresh media to allow expression of the virally-encoded cDNA[21]. For general ER-to-Golgi protein trafficking experiments, NRVMs were infected with CellLight Golgi-RFP Bacmam 2.0 (ThermoFisher Scientific, #c10593) that contains a fusion construct of the Golgi protein N-acetylgalactosaminyltransferase with RFP (GalNacT2-RFP) 24 h after AdThbs3 or AdThbs4 infection. For Golgi-to-membrane protein trafficking experiments, NRVMs were infected with adenovirus harboring the temperature sensitive VSVG-eGFP and incubated at 40 °C for 24 h to retain the VSVG-eGFP in the ER, 24 h after infection with AdThbs3 or AdThbs4. FRAP and iFRAP imaging were performed as described above.

**Electron microscopy**. Hearts of anesthetized mice were perfused with relaxing buffer (0.15% sucrose, 5% dextrose, 10 mM KCl in PBS) for 3 min and then perfused with fixation buffer (1% PFA, 2% glutaraldehyde in 100 mM sodium cacodylate pH 7.4), then fixed overnight in fixation buffer and post-fixed in 1% OsO$_4$ for 2 h. Ultrathin sections of all tissues were counterstained with uranyl acetate and lead salts. Images were obtained using a 7600 transmission electron microscope (Hitachi) connected to a digital camera (AMT, Biosprint16).

**Histological analysis and immunohistochemistry**. Heart tissue was fixed overnight in 10% formaldehyde, dehydrated in ethanol and paraffin embedded. Tissue was sectioned at 5 μm and stained with either H&E or Masson's trichrome. Images were captured with a stereo microscope (Leica Microsystems, #M165FC) equipped with a digital camera (Leica Microsystems, #DFC310 FX) and the Leica Application Suite. Interstitial fibrosis was quantified using Photoshop software as the percentage of the trichrome-stained area (blue) over the total tissue area[70]. For co-labeling of tissue sections, 5 μm sections were rehydrated and heated in 1× antigen retrieval CITRA (BioGenex, # HK086-9K). Analysis of real time Thbs3 expression using the β-galactosidase reporter mouse line was performed using frozen sections. Hearts embedded in O.C.T compound (VWR, #25608-930), frozen in liquid nitrogen and tissue sections were cut at a thickness of 7 μm. Heart sections were permeabilized for 2 min in 0.1% triton/PBS and incubated for 1 h at room temperature in blocking buffer (PBS, 5% goat serum, 2% BSA). Primary antibody incubations were performed overnight at 4 °C: integrin β1D (EMD Millipore, #MAB1900 at 1:100), PDI (Abcam, #ab2792 at 1:100), Thbs3 (Proteintech, #19727-1-AP at 1:100). Desmin (Sigma, #D1033 at 1:200), vimentin (Abcam, #ab45939 at 1:200), β-glactosidase (Abcam, #9361 at 1:100) and isolectin B4 (Vector Laboratories, #B-1201). Appropriate Alexa Fluor-488 (ThermoFisher Scientific, Goat anti-Mouse IgG, #A11029; goat anti-rabbit IgG, #A11008) and Alexa Fluor-568 (ThermoFisher Scientific, goat anti-mouse IgG #A11031; Goat anti-Rabbit IgG #A11036) secondary antibodies at 1:500 in blocking buffer were applied for 1 h at room temperature and subsequently for 10 min with DAPI (4′,6-diamidino-2-phenylindole) nuclear DNA stain at 1:10,000 (ThermoFisher Scientific, #D1306). Staining with wheat germ agglutinin conjugated to FITC (Lectin from Triticum vulgaris, Millipore Sigma, #L4895 at 1:100) was done for 1 h at room temperature to visualize the membranes.

NRVMs were infected with Adβgal control, AdThbs3, AdThbs3 RGD, or AdThbs4 adenoviruses. Cells were grown in Medium 199/EBSS (ThermoFisher Scientific, #SH30253FS) supplemented with 2% bovine growth serum

(ThermoFisher Scientific, # SH3054103) and 1% penicillin-streptomycin (Cellgro, #30-0002-CI) for 48 h. Then, cells were washed in PBS, permeabilized for 2 min in 0.1% triton/PBS and incubated for 1 h at room temperature in blocking buffer (PBS, 5% goat serum, 2% BSA). Primary antibody incubations were performed overnight at 4 °C, Thbs3 (Proteintech, #19727-1-AP at 1:100) and Thbs4 (Santa Cruz Biotechnology, #sc-7657 at 1:100), followed by incubation with secondary antibody conjugated to Alexa Fluor-488 (ThermoFisher Scientific, Goat anti-Rabbit IgG, #A11008) for 1 h at room temperature and subsequently for 10 min with DAPI (4′,6-diamidino-2-phenylindole) nuclear DNA stain at 1:10,000 (ThermoFisher Scientific, #D1306).

All sections were mounted in Vectashield Hard Set (Vector Laboratories, #H-1400) to prevent photo bleaching and visualized using a Nikon A1+ confocal laser microscope system (Nikon Instruments) equipped with 40x H$_2$O objective (Nikon Instruments, NA = 1.15). All imaging was done under identical conditions using NIS Elements Advanced Research microscope imaging software (Nikon Instruments).

**RNA isolation and quantitative reverse-transcriptase PCR**. RNA was isolated from hearts using RNeasy Fibrous Tissue Kit (QIAGEN, #74704) according to the manufacturer's instructions. Synthesis of cDNA was performed using Superscript III First Strand Kit (Invitrogen, #18080-051) with random hexamer primers according to the manufacturer's instructions. Quantitative real-time PCR was performed using SsoAdvanced SYBR Green (Bio-Rad, #6090). Mouse Thbs3 expression was analyzed (forward 5′-tcgccaagacaacacacg-3′, reverse 5′-catg-gactttggccatcttcc-3′) and Rpl13 (forward 5′-gccggactccctacaagc-3′, reverse 5′-gcttcagtatcatgccattcc-3′) was used as a reference for relative quantification.

**EBD uptake experiments**. Eight week-old mice were intraperitoneally injected with Evan's blue dye (EBD, 10 mg/ml; 0.1 ml per 10 g body weight (Sigma Aldrich #E2129)), 24 h later mice were injected (i.p.) with isoproterenol (300 mg/kg)[31]. Mice were sacrificed 24 h later and hearts were harvested, embedded in Tissue-Tek O.C.T. (VWR, #25608-930) and frozen in liquid nitrogen. Tissue sections were cut at a thickness of 7 μm, air-dried, washed in PBS and stained with wheat germ agglutinin conjugated to FITC (Lectin from Triticum vulgaris, Millipore Sigma, #L4895) for 1 h at room temperature to visualize the membranes. Images were taken on a Citation 5 image scanner (Biotek), equipped with GFP and Texas Red filter sets, and analyzed automatically to determine the percentage of EBD-positive area as red staining.

**Isolation of adult cardiac myocytes and Ca$^{2+}$ measurements**. To isolate adult mouse ventricular cardiac myocytes, the heart was rapidly excised and placed in Tyrode's solution (130 mM NaCl, 4 mM KCl, 2 mM CaCl$_2$, 1 mM MgCl$_2$, 10 mM glucose, 10 mM HEPES, pH 7.4) in the presence of 10 mM 2,3-butanedione monoxime (BDM, Millipore Sigma, #B0753), 5 mM taurine (buffer A) and gassed with 95% O$_2$ 5% CO$_2$. All solutions were equilibrated with 95% O$_2$ 5% CO$_2$ for 20 min before use. The heart was perfused in retrograde direction with buffer A for 5 min, then 1 mg/ml collagenase type II (Worthington, Freehold, NJ, USA) and 0.08 mg/ml protease type XIV (Sigma) was added to buffer A at 37 °C. After 2 min of enzyme perfusion, 50 μM Ca$^{2+}$ was added to the enzyme solution. The heart was then perfused for an additional 12 min or until flow rate surpassed pre-enzyme flow rate. After perfusion, the ventricles were separated from the atria and minced and filtered to single cardiac myocytes for analysis Cardiac myocytes were loaded with 2 μM Fura-2 acetoxymethyl ester (ThermoFisher Scientific, # F-1221) for 15 min and placed in Tyrode's solution. Caffeine was added acutely to measure sarcoplasmic reticulum Ca$^{2+}$ load. The Fura-2 fluorescence ratio was determined using a delta scan dual-beam spectrofluorophotometer (Horiba, #D-104) operated at an emission wavelength of 510 nm and excitation wavelengths of 340 and 380 nm. The stimulation frequency for Ca$^{2+}$ transient measurements was 1 Hz. Ca$^{2+}$ traces were processed using a Savitzky-Golay filter and baseline Ca$^{2+}$ levels, transient amplitude, and Ca$^{2+}$ decay kinetics were analyzed using Felix 1.1 and Ion Wizard software (IonOptix).

**Luciferase folding assays**. Chaperone activity was measured using firefly luciferase full-length protein (Abcam, #ab100961) and native luciferase activity was set to 100% activity. Luciferase (1 mg/ml) was diluted 4-fold in denaturation buffer (25 mM Hepes, pH 7.4, 50 mM KCl, 5 mM MgCl$_2$, 6 M guanidine-HCl, 5 mM dithiothreitol). The denaturation reaction was carried out at room temperature for 40 min. Denatured luciferase was diluted 125 fold in refolding buffer (25 mM Hepes, pH 7.4, 50 mM KCl, 5 mM MgCl$_2$, 1 mM ATP) supplemented with 2 μM BSA or 2 μM of the indicated chaperone protein and incubated at room temperature for 60 min (recombinant heat shock protein 70 (Enzo Life Sciences, #ADI-NSP-555-D), recombinant Thbs4 (R&D Systems #7860-TH-050), recombinant Nell2 (Aviva Systems Biology #OPPB00457), bovine serum albumin (BSA, ThermoFisher Scientific, #BP1600), recombinant Thbs3 (R&D Systems #8390-TH-050)). Reactions were carried out in triplicates, diluted 2-fold in luciferase assay buffer (Promega, #E1500) and luciferase activity was measured using a luminescence reader (Perkin Elmer, #2030 Victor X Light).

**Statistics and reproducibility**. Results are presented in all cases as mean ± SEM. Statistical analysis was performed using Graphpad Prism 6 (Graphpad Software). Individual variable comparison was analyzed using two-tailed $t$-test (HW/BW weight, LW/BW weight, calcium measurements, LVIDD, LVIDS, fibrotic area, protein intensities), $P$-values less than 0.05 were considered significant. Comparison of multiple groups was performed using one-way analysis of variance (ANOVA) with Turkey multiple comparisons test (HW/BW weight, FS, LVIDD, EBD positive area, protein intensities), $P$-values less than 0.05 were considered significant. Comparison of multiple groups at multiple time points was performed using two-way analysis of variance (ANOVA) with Turkey multiple comparisons test (trafficking experiments), $P$-values less than 0.05 were considered significant. Comparison of survival curves was performed with log-rank (Mantel-Cox) test, $P$-values less than 0.05 were considered significant

**EBD quantification**. Image acquisition and analysis was done using Gen5 data analysis software (Biotek). WGA positive area of heart tissue sections was determined to quantify heart tissue area. EBD positive area was measured and the percentage of EBD positive area of the WGA positive area was calculated. One-way analysis of variance (ANOVA) with Turkey multiple comparisons test was used and $P$-values less than 0.05 were considered significant.

## Data availability

All original data underlying selected data shown in the figures and supplementary figures are available from the corresponding author upon reasonable request. A reporting summary for this Article is available as a Supplementary Information file.

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

## Acknowledgements

We thank Dr. J.T. Hecht for providing *Thbs3* and *Thbs5* gene-deleted animals and Dr. T. Kyriakides for providing *Thbs2* gene-deleted animals. This work was supported by a grant from the German research foundation (SCHI 1290/1-1) to T.G.S., NIH (2R01HL105924) and the Howard Hughes Medical Institutes to J.D.M., and NIH (1R01HL127806 and 1RO1HL115933) to R.S.R.

## Author contributions

T.G.S., J.D.M. conceptualized the study; T.G.S., D.V., J.D.M. designed the experiments; T.G.S., D.V., A.V. R.N.C., M.J.B., H.K., A.T., J.K., M.A.S performed experiments; T.G.S., D.V., A.V., M.J.B., H.K., J.K. analyzed data; R.S.R. contributed experimental animals; T.G.S., M.J.B., M.M., R.S.R., and J.D.M. wrote or edited the manuscript.

## Additional information

**Competing interests:** The authors declare no competing interests.

