## [Peer Review File · Nature Communications]

Reviewers' comments:

Reviewer #1 (Remarks to the Author):

The study reports for the first time effects of Thrombospondin-3 (Thbs3) in myocardial injury. The authors report that: a) Thbs3 is upregulated in the genetic and pressure overload models of cardiomyopathy, b) cardiomyocyte-specific Thbs3 overexpression has no effects on cardiac function and histology, but exacerbates the cardiomyopathy induced through pressure overload, c) global loss of Thbs3 attenuates pressure overload-induced hypertrophy and fibrosis, d) mechanistically, Thbs3 downmodulated intracellular integrin expression, preserving sarcolemmal stability following injury.

General comment:

This is an interesting study, reporting a novel mechanism on the effects of the thrombospondins in the myocardium. Practically nothing is currently known regarding the role of Thbs3 in heart disease. The mechanism explaining the contrasting effects of Thbs3 and Thbs4 on integrin expression is of particular interest. The main weakness is the unclear pathophysiologic significance of the findings, considering that most of the experiments were performed in artificial models of overexpression. The following major concerns need to be addressed:

Major comments:

1. A good part of the study is based on artificial models of overexpression with limited pathophysiological relevance. The relevance of the cardiomyocyte-specific TSP-3 overexpression experiments seems relatively limited. Data on expression and localization of Thbs3 in pathophysiologically relevant models are weak and limited. Localization of TSP3 in cardiomyocytes is studied only in the (artificial) activated calcineurin overexpression model of cardiomyopathy. What is the cellular localization of TSP-3 following TAC? Is TSP3 predominantly expressed in cardiomyocytes? Is it also secreted and deposited in the matrix?
2. The notion that Thbs3 is upregulated following TAC is based on representative WB images showing a very weak band. A time course of expression and quantitative data are needed.
3. Clearly, the pathophysiologically significant findings of the study relate to the effects of TSP-3 loss in the model of cardiac pressure overload. This direction should be expanded. The basis for the observed protection needs to be explored. Considering the use of a global KO line, what is the cellular basis for the observed effects? Do they reflect effects on cardiomyocytes? Are there direct anti-fibrotic actions? Are there effects on macrophage phenotype?

4. What is the mechanism for TSP-3 induction in pressure-overloaded hearts? Does mechanosensitive signaling induce TSP-3 expression in cardiomyocytes?

5. The rationale for the exclusive focus on effects of TSPs on integrin levels is unclear.

6. Do Thbs3 KO mice have any baseline defects?

7. The abstract requires minor revisions and clarifications:

“Deletion of just Thbs3 in mice also enhances integrin membrane expression and membrane stability during injury, preserving cardiac function”. Please specify the type of injury

“Transgene-mediated overexpression of integrin $\alpha7\beta1D$ in the heart ameliorates the detrimental effects of Thbs3 by augmenting integrin function and sarcolemmal stability.” Why this specific integrin? In what context (what type of injury) does the integrin ameliorate the detrimental effects of Thbs3?

8. The title is somewhat misleading, as Thbs3 does not promote cardiomyopathy in the absence of injury. Please revise.

Reviewer #2 (Remarks to the Author):

This manuscript presents novel genetic evidence that thrombospondin-3 (Thbs3) expression promotes cardiomyopathy, which was unexpected given the opposite activity of thrombospondin-4. The genetic data are strong and utilize both loss and over-expression alone and in the context of deleting other thrombospondin genes. The authors propose a mechanistic basis for this divergence based on differential regulation of integrin trafficking. The data supporting this mechanism are less convincing and require additional controls.

1. Fig 3 requires quantitative analysis to assess statistics. Also, additional controls are needed. Does the abundance of integrins differ in whole tissue lysates from Thbs3DTA versus controls? This is important to assess whether the missing integrins selectively accumulate in an intracellular compartment or are expressed less overall in the Thbs3 DTG hearts.

2. Fig 3b shows Thbs3 in a microsomal prep from cultured rat ventricular myocytes, but this can't be interpreted in isolation. Showing more Thbs3 in microsomes after infecting with Thbs3-Ad is a trivial result that may have not relevance to the heart studies. Does over-expressing Thbs3 in these cells alter cell surface versus intracellular levels of the same integrins that show decreases in Fig 3a? The

authors draw the conclusion that “Thbs2 acts as an intracellular repressor of integrin membrane trafficking in the heart and thereby destabilizes the sarcolemma.” This is one hypothesis that is consistent with the results shown, but the data presented do not specifically test this hypothesis or exclude any alternative hypotheses.

3. Supplemental fig 2b is not convincing. Comparing single isolated sections does not establish a significant loss of membrane integrity in the Thbs3 DTG heart after TAC. Because TAC is the protocol relevant to interpreting the other main figures, rather than the isoproterenol experiment shown in Fig 3d, this result needs to be well documented in the context of TAC.

4. Fig 3 clearly documents protection of stressed Thbs3^{-/-} hearts, but the mechanistic data is weak. ItgB1D is obviously up in sarcolemma extracts from only one of the two hearts, so the conclusion in the results seems overstated. Again, statistics are needed, and control blots to determine whether ItgA5 and ItgA7 are up only in sarcolemma or also in bulk heart tissue. Why do integrins also appear to be up in one of the Thbs3^{-/-} sham controls?

5. Why were Thbs1/2/3/4/5^{-/-} mice not included in Fig 5f and 5g? That is relevant control to provide a mechanistic correlate for the phenotypes in Fig 5. It is notable that Thbs3^{-/-} show the same inconsistent ItgB1D response in 5g as in Fig 4g.

6. Fig 7 addresses some of the concerns raised regarding effects of Thbs3 on integrin localization in Fig 3. However, it is unclear whether the modest decreases in surface ItgA5 and ItgB1D are significant relative to the Adbgal control or reproducible. Replicate experiments need to be presented in the supplement and statistical analysis performed.

7. Given the evidence that exogenous Thbs1 was not effective in earlier experiments, how can the proposed mechanism explain the uptake data in Fig 7f. Does exogenous Thbs3 enhance uptake of surface integrins?

8. The reversed activity of AdThbs3RGD in integrin trafficking is interesting, but the integrin localization and uptake data are limited to a RGD-binding integrin. Prior studies indicated that the ItgA7/B1 interaction with its major ligand laminin is not RGD dependent. Do AdThbs3 and AdThbs3RGD have differential effects on surface localization and uptake of ItgA7?

Reviewer #3 (Remarks to the Author):

Summary: The current manuscript utilizes multiple mouse models to investigate the influence of Thbs on cell surface expression of integrins, thereby regulating the heart's susceptibility to TAC/ISO treatments and membrane stability. The physiology data is strong and the extent of EBD uptake into the Thbs3 DTG hearts after isoproterenol treatment is remarkable (Fig. 3 d). However, the manuscript does not convincingly demonstrate alterations in integrin cell surface expression with sufficient rigor. The manuscript contributes to the concept of integrin shedding and its role in cardiomyocyte remodeling, which is interesting and important. It is the sense of this reviewer that, while these findings are relevant, they may be more appropriate for a cardiac focused journal.

Specific Comments:

The concept of integrin shedding as a component of the remodeling process is appealing, especially in the context of the data presented. It is the opinion of this reviewer that this component, now found in the discussion, could also appear earlier in the manuscript.

Electron micrographs to depict increased membrane vesicles in Thbs3 DTG hearts are blurry and not convincing based on the limitations of the small field provided. It also appears that the mitochondria morphology may be altered. Could the authors comment on this?

In general, the rigor of the immunoblotting data on cardiac sarcolemma membrane preps as an approach to show changes in membrane expression of proteins is insufficient in its current execution. This data is central to the argument that Thbs affect cell surface expression of integrins. Quantitation (including n-values) of protein intensities are needed. Validation of the membrane enrichment is lacking (are there non-membrane associated proteins in the prep? What is the extent of membrane enrichment? Do the preps contain internal membranes or cell surface membranes?). There are many approaches to increase the rigor of these fairly standard biochemical experiments. Standard MW markers are needed. An additional approach, such as localization of integrins in the heart, should be provided to support that Thbs affects integrin membrane localization. While this is shown for integrin beta1 in figure 5f, the study is limited. The surface biotinylation of NRVMs treated with Ad expressing Thbs is a valid approach to addressing cell surface expression (Figure 7f). However, the data in its current form isn't convincing.

Figure 3a: Which laminin polypeptide is shown in the panel? Please use correct symbols for proteins. What does the arrow refer to on the panel? Can the author comment on how they conclude that the upper Thbs3 band is non-specific? Were antigen competition studies performed? What is the evidence that the preps represent the cell surface? While Cacna1c serves as a positive control for sarcolemma protein content, a negative marker should be provided (actin or other). Do the preps contain ER/Golgi membranes as well?

Figure 3b: The IP experiments lack rigor in their current state. Input microsomal fractions should be shown and reciprocal IPs with Thbs3 antibody showing pull down of integrin beta1 would strengthen the evidence of protein-protein interactions. Can the authors comment on the two protein bands observed with the integrin beta1 antibody?

Figure 4g: Are the blots for sham and TAC treated samples from the same exposure? In other words, is it possible to compare the sham and TAC samples? The data supporting differences in sarcolemma localization of integrins upon Thbs3 knockout is weak in its current form and would be strengthened by follow up experiments. These could include localization on cardiac sections using integrin antibodies or cell surface labeling. Controls to support the stringency of the sarcolemma enrichment from heart samples is lacking. What are the multiple bands in integrin beta1D immunoblots?

The data in figure 5f reveal integrin beta1D expression at the sarcolemma in WT and Thbs3 nulls after TAC. There is a notable increase in integrin expression at the sarcolemma in TAC treated THBS3-nulls relative to WT hearts. However, the immunoblot data from Figure 4g suggests that there are no differences in integrin beta1D abundance in microsomal preps from these same genotypes. It is unclear what the differences are between blots shown in Figure 5g and 4g as they both appear to show integrin beta1, integrin alpha 5 and integrin alpha 7 staining for the same TAC treated WT and Thbs3-nulls.

Response to Reviewers' comments:

Reviewer #1 (Remarks to the Author):

General comment:

The main weakness is the unclear pathophysiologic significance of the findings, considering that most of the experiments were performed in artificial models of overexpression.

While our paper does present data with Thbs3 overexpression in the hearts of mice, we believe that this concern is overstated. We actually spent over 2 years generating five-way knockout mice for all five thrombospondin genes and compared them to four-way knockout mice that only contain endogenous Thbs3, as a physiologic way to show the function of remaining Thbs3 without overexpression. I cannot over emphasize what an extreme measure this experimental approach represented for my laboratory, as it literally took us 2 years just to generate 4-way and 5-way KO mice as an approach to investigate the physiologic function of Thbs3 without overexpression. An entire figure is dedicated to these data, as well as supplemental data and panels of data in another figure. These data are clearly a major feature of the paper and I worry that the reviewer did not fully appreciate their significance. More than this, we also show a great deal of data with Thbs3 KO mice in addition to the 4 way KO mice, and while the 4-way KO mice that only have Thbs3 are more susceptible to heart disease with pathologic stimulation, just like the Thbs3 overexpressing mice, the Thbs3 single KO mice are protected from heart disease. KO of Thb3 is also considered a physiologically relevant approach towards understanding gene functionality in vivo. Hence, we believe that the entire experimental design is profoundly physiologically relevant, and more importantly, the results with these 2 loss-of-function approaches match the results obtained with the Thbs3 overexpressing transgenic mice to paint a highly consistent picture. Finally, we also even mutate Thbs3 to now contain an RGD domain that is present within all other Thbs family members, and this mutant form of Thbs3 now reverses the disease predisposing effects of integrin clearance from the membrane when overexpressed, hence all of our data are verified on many levels and we strongly believe that our results reflect the true biology of Thbs3.

Major comments:

1. A good part of the study is based on artificial models of overexpression with limited pathophysiological relevance. The relevance of the cardiomyocyte-specific TSP-3 overexpression experiments seems relatively limited. Data on expression and localization of Thbs3 in pathophysiologically relevant models are weak and limited. Localization of TSP3 in cardiomyocytes is studied only in the (artificial) activated calcineurin overexpression model of cardiomyopathy. What is the cellular localization of TSP-3 following TAC? Is TSP3 predominantly expressed in cardiomyocytes? Is it also secreted and deposited in the matrix?

See our main rebuttal above as to the physiologic relevance of entire experimental approach and why we strongly believe that our results are showing us the true function of Thbs in the heart during pathophysiological stimulation. However, we agree that our expression data and

immunolocalization data needed reinforcement. In the revised paper we now more thoroughly investigated the expression and localization of Thbs3 following TAC (**New Figure 1a, b**) and added qPCR expression data after TAC (**new Supplemental Figure 1b**). In addition immunohistochemically analysis of Thbs3 localization after TAC was performed and shows Thbs3 in cardiomyocytes (**new Figure 1b**). We have also added new data to the paper in which we used Thbs3-LacZ knock-in mice to show expression of this allele from the endogenous Thbs3 locus, and that expression is observed only with stress stimulation, but that expression is within myocytes, capillaries and fibroblasts (**new Supplemental Figure 1c**). As we already showed that we can detect secreted Thbs3 in vitro and we have previously observed that Thbs3 protein is also taken up by cardiomyocytes (**Supplemental Figure 7b and uncropped gel image file**). However we are unable to detect Thbs3 being deposited in the extracellular matrix in vivo. We have observed this same set of data for Thbs4 in both heart and skeletal muscle in separate papers (Lynch et al., 2012, Vanhoutte et al., 2016). We are extremely confident in the broader result that Thbs3 resides within the cell (with antibodies verified in KO mouse samples), and while it traffics with cargo to the membrane and is secreted, it does not reside in the ECM and is recycled back into the cell. However, we have added more data to better show the expression characteristics of endogenous Thbs3 in the revised paper, as requested by this reviewer

2. The notion that Thbs3 is upregulated following TAC is based on representative WB images showing a very weak band. A time course of expression and quantitative data are needed.

In agreement with the reviewers remark we have repeated the expression analysis in the heart failure models (**new Supplemental Figure 1a**). We also added Protein quantifications to the analysis and included a time course analysis of Thbs3 expression after TAC (**new Figure 1a**). We have also added a time course analysis of Thbs3 mRNA levels in the heart after TAC (**new Supplemental Figure 1b**)

3. Clearly, the pathophysiologically significant findings of the study relate to the effects of TSP-3 loss in the model of cardiac pressure overload. This direction should be expanded. The basis for the observed protection needs to be explored. Considering the use of a global KO line, what is the cellular basis for the observed effects? Do they reflect effects on cardiomyocytes? Are there direct anti-fibrotic actions? Are there effects on macrophage phenotype?

We thank the reviewer for raising these concerns and we have added new data to the paper to better make our case. As discussed above, we extended our analysis of Thbs3 localization following TAC and can show that it is expressed in cardiomyocytes, but also fibroblasts and capillaries and vessels, which is consistent with its known ubiquitous expression pattern after injury (**New Figure 1b and Supplemental Figure 1c**). However, our data suggest that it is Thbs3 functioning within the cardiomyocyte that is the relevant disease predisposing celltype underlying our results. The Thbs3 overexpressor mouse is cardiac myocyte restricted, so all primary effects are due to its expression in this cell type. Comparing results with this approach to data obtained in the Thbs1/2/4/5 quadruple KO mice that only have Thbs3 remaining in all

cell types, show that both models have the same disease predisposing effect on membrane instability in the heart and loss of integrins from the sarcolemma. More than this, we devote an entire figure to a rescue experiment using myocyte-specific overexpression of $\beta 1/\alpha 7$ integrin transgenic mice and this rescues the defective membrane phenotype and disease predisposition associated with Thbs3 (**Figure 6**). Finally, we did observe an effect on the fibrotic response in Thbs3 KO mice after TAC stimulation (**Figure 4F**), as well as less immune cell infiltration in the heart after TAC stimulation compared with WT controls (**see Figure here for reviewer**), although we believe that these 2 effects are due to less ongoing membrane damage to the myocytes and resulting secondary inflammatory. However, we do admit that it is possible that loss of Thbs3 produces subtle effects on fibroblasts in the heart, although Thbs3 KO mice do not show other defects or alterations in their immune response.

4. What is the mechanism for TSP-3 induction in pressure-overloaded hearts? Does mechanosensitive signaling induce TSP-3 expression in cardiomyocytes?

The reviewer raises an important question about the mechanism of Thbs3 induction after TAC. Thbs3 is upregulated transcriptionally and it seems to be independent of the type of injury or the cell type. Several publications show induction of Thbs3 mRNA expression in various cell types with stimulation or injury (as listed below). We also added an analysis of mRNA induction over a time course of TAC in the heart (**new Supplemental Figure 1b**)

(Myofibroblasts: *Matrix Biol.* 2018 Jan;65:59-74. doi: 10.1016/j.matbio.2017.07.005.

Identification of a myofibroblast-specific expression signature in skin wounds. Bergmeier V. et al.;

Bone marrow endothelial cells: *International Journal of Molecular Medicine*, 32(5), 1204-14, 2013 Whole genome expression profiling and screening for differentially expressed cytokine genes in human bone marrow endothelial cells treated with humoral inhibitors in liver cirrhosis. Gao B et al. doi.org/10.3892/ijmm.2013.1495;

Brain: *Neuroimmunomodulation*. 2007;14(1):46-56. Differential expression of extracellular matrix and adhesion molecule genes in the brain of juvenile versus adult mice in responses to intracerebroventricular administration of IL-1. Ching S et al.; Fallopian tube *Reprod Biol Endocrinol.* 2012 Aug 16;10:56. doi: 10.1186/1477-7827-10-56.

Fallopian tubes: Differential expression of extracellular matrix components in the Fallopian tubes throughout the menstrual cycle. Diaz PS et al.;

Chronic myeloid leukemia: *Differentiation*. 2008 Oct;76(8):908-22. doi: 10.1111/j.1432-0436.2008.00270.x. Phenotypic and gene expression diversity of malignant cells in human blast crisis chronic myeloid leukemia. Simanovsky M et al.;

Osteosarcoma: *BMC Cancer*. 2006 Oct 5;6:237. Effects of THBS3, SPARC and SPP1 expression on biological behavior and survival in patients with osteosarcoma. Dalla-Torre CA et al.)

Furthermore we analyzed Thbs3 expression in skin punch wounds at different time points and found significant upregulation of Thbs3 expression, again suggesting transcriptional induction of Thbs3 independent of mechanosensing (**data included here for this reviewer**).

5. The rationale for the exclusive focus on effects of TSPs on integrin levels is unclear.

We thank the reviewer for this remark and we will attempt to clarify this issue. We believe that our paper is very logical in progression in culminating with a common biologic pathway implicating integrin regulation at the sarcolemma. First, our previous work showed that Thbs4 overexpression or deletion in skeletal muscle profoundly affected the stability of the sarcolemmal membrane, which directly affected muscular dystrophy phenotypes (Vanhoutte et al., 2016). We specifically showed that Thbs4 overexpression in skeletal muscle profoundly upregulated select integrins at the sarcolemma, as well as components of the DGC through enhanced intracellular vesicular trafficking of these complexes to the plasma membrane, while loss of Thbs4 weakened the cell membrane and lead to spontaneous muscular dystrophy in mice (Vanhoutte et al., 2016). In that same paper we also showed that Thbs4 in *Drosophila* also impacted skeletal muscle membrane stability and content of fly β PS integrin content in the plasma membrane (Vanhoutte et al., 2016). When we generated Thbs3 overexpressing transgenic mice we expected to observe the same phenotype of protection in mice and increased membrane stability, as Thbs3 and Thbs4 are both pentameric family members and somewhat closely related. However, we observed the exact opposite phenotype at the level of the plasma membrane, whereby Thbs3 overexpression caused membrane instability and dramatically reduced integrin content at the membrane. This result alone suggested that the underlying mechanism for affecting the membrane was 2 sides of the same coin; altering integrin surface activity as a primary mechanism in play. We were pointed in this direction years ago because our work on muscular dystrophy is inherently related to membrane fragility as the unifying mechanism for all disease, and because the membrane attachment complexes between the DGC and integrins is the primary system in place to protect skeletal muscle sarcolemmal membrane integrity. However, we were also aware of past literature in which the Thbs genes were shown to function as effectors of integrin activity, which was also shown to be the exclusive mechanism of action for the single *Drosophila* Thbs gene, indicating evolutionary conservation of the most central regulatory effect of the Thbs family potentially operating in mammals.

(Thrombospondin-mediated adhesion is essential for the formation of the myotendinous junction in *Drosophila*. Subramanian A et al.; *Mech Dev*. 2007 Jul;124(6):463-75.

AlphaPS2 integrin-mediated muscle attachment in *Drosophila* requires the ECM protein Thrombospondin. Chanana B et al.; *Circ Res*. 2007 May 11;100(9):1308-16.

Interaction of alpha9beta1 integrin with thrombospondin-1 promotes angiogenesis. Staniszewska I et al.; *Development*. 2007 Apr;134(7):1269-78. *J Med Chem*. 2006 Oct 19;49(21):6324-33.

Conformational analysis of an alpha3beta1 integrin-binding peptide from thrombospondin-1: implications for antiangiogenic drug design. Furrer J et al.; *Cancer Res*. 2000 Jan 15;60(2):457-66.

Thrombospondin-1 promotes alpha3beta1 integrin-mediated adhesion and neurite-like outgrowth and inhibits proliferation of small cell lung carcinoma cells. Guo N et al.).

These papers listed above are just a small sampling of the literature suggesting that the entire Thbs gene family may have evolved as means of altering integrin activity in the plasma membrane during times of healing or injury (or development as in the fly). Comparing the effects of Thbs3 and Thbs4 on cell surface receptors revealed an antithetic effect whereby Thbs4 specifically augmented integrin and DGC attachment complexes while Thbs3 reduced these complexes at the plasma membrane, both of which effects are likely necessary for healing whereby cells move and reattach as an area is repaired (see **Supplemental Figure 8**). Finally, while Thbs4 also increased the membrane content of DGC components, Thb3 did not effect DGC membrane levels, as the integrin modulation mechanism was far more prevalent. Overall, these considerations underlied our mechanistic focus on the integrins.

6. Do Thbs3 KO mice have any baseline defects?

The reviewer raises an important question about a possible baseline phenotype of Thbs3 KO mice. The initial characterization of Thbs3 KO mice (*Mol Cell Biol*. 2005 Jul;25(13):5599-606. Mice with a disruption of the thrombospondin 3 gene differ in geometric and biomechanical properties of bone and have accelerated development of the femoral head. Hankenson KD et al.) reported no overt baseline defects except for altered endochondral bone formation, with accelerated ossification of the femoral head. In agreement with this report, we did not observe any cardiac defects in Thbs3 KO mice. Sham-operated control animals were analyzed at 5 months of age and showed no differences in HW/BW ratio, heart function and geometry when compared to Wild type control animals (see **Fig 4**).

7. The abstract requires minor revisions and clarifications:

We apologize to the reviewer and we agree that these passages were not very clear. In agreement with the reviewers comment we have now corrected the passages in the abstract as follows:

“Deletion of just Thbs3 in mice also enhances integrin membrane expression and membrane stability during **pressure overload mediated cardiac** injury, preserving cardiac function

“Transgene-mediated overexpression of $\beta 1/\alpha 7$ integrin, the dominant integrin in the normal adult heart, ameliorates the detrimental effects of Thbs3 with isoproterenol infusion by augmenting integrin function and sarcolemmal stability.” Why this specific integrin? In what context (what type of injury) does the integrin ameliorate the detrimental effects of Thbs3?

8. The title is somewhat misleading, as Thbs3 does not promote cardiomyopathy in the absence of injury. Please revise.

We have changed the title to: "*Thrombospondin-3 augments injury-induced cardiomyopathy by intracellular integrin inhibition and sarcolemmal instability*"

Reviewer #2 (Remarks to the Author):

1. Fig 3 requires quantitative analysis to assess statistics. Also, additional controls are needed. Does the abundance of integrins differ in whole tissue lysates from Thbs3DTA versus controls? This is important to assess whether the missing integrins selectively accumulate in an intracellular compartment or are expressed less overall in the Thbs3 DTG hearts.

We thank the reviewers for these valuable remarks and added quantitative statistical analysis for Figure 3a in **new Supplemental Figure 3**. We also added integrin protein analysis of whole tissue lysates from tTA control and Thbs3 DTG hearts which we are showing the reviewer here (below). These data show that total integrin content in whole cell heart tissue extracts does not change, and that the alterations in the integrin levels shown in **Figure 3a** is

specific to cell membrane occupancy due to regulation by Thbs3/4 proteins on residence of these identified integrins, which we have clarified on **page 8**. Moreover, we were unable to show intracellular accumulation of integrin staining in Thbs3 DTG heart sections (**Supplemental Figure 3l-n**). This might be due to specificity of the antibody in this assay for mature, cell surface integrins in contrast to immature intracellular integrins, or due to a concentration effect that we discuss again below in response to the same concern reiterated again, whereby only at the plasma membrane do the integrins achieve a high enough concentration to give strong antibody reactivity.

2. Fig 3b shows Thbs3 in a microsomal prep from cultured rat ventricular myocytes, but this can't be interpreted in isolation. Showing more Thbs3 in microsomes after infecting with Thbs3-Ad is a trivial result that may have not relevance to the heart studies. Does over-expressing Thbs3 in these cells alter cell surface versus intracellular levels of the same integrins that show decreases in Fig 3a? The authors draw the conclusion that “Thbs2 acts as an intracellular repressor of integrin membrane trafficking in the heart and thereby destabilizes the sarcolemma.” This is one hypothesis that is consistent with the results shown, but the data presented do not specifically test this hypothesis or exclude any alternative hypotheses.

We agree with the reviewer's opinion that more Thbs3 in microsomes after infecting with Thbs3-adenovirus is a trivial result, but this was not the important feature of the experiment as shown. The experiment presented in Figure 3b is an immunoprecipitation of intact microsomes using an integrin $\beta 1$ antibody that recognizes the c-terminal (cytoplasmic) domain of integrin $\beta 1$. This allows the isolation of integrin $\beta 1$ containing vesicles. The experiment shows that Thbs3 localizes to the same vesicles as integrin $\beta 1$. We apologize to the reviewer for the unclear description of the experiment presented in Figure 3b, which we have now revised to make it more clear in the results section. We have also added control blots to show the input of these proteins in the given fraction (**new Figure 3b**).

We also agree that stating the conclusion that Thbs3 acts as an intracellular repressor of integrin trafficking in the results section of Figure 3 is premature and we have deleted this statement at this point in the manuscript.

3. Supplemental fig 2b is not convincing. Comparing single isolated sections does not establish a significant loss of membrane integrity in the Thbs3 DTG heart after TAC. Because TAC is the protocol relevant to interpreting the other main figures, rather than the isoproterenol experiment shown in Fig 3d, this result needs to be well documented in the context of TAC.

The reviewer raises an important point about the significance of lost membrane integrity in Thbs3 DTG hearts after TAC. We have revised the paper and performed a quantitative analysis of EBD uptake after TAC that indeed shows a significant increase in EBD positive cardiomyocytes after TAC (**new Figure 3g**). We have also moved these data with the quantitation from the supplemental figure to a primary figure (**revised Figure 3e-g**).

4. Fig 3 clearly documents protection of stressed Thbs3^{-/-} hearts, but the mechanistic data is weak. ItgB1D is obviously up in sarcolemma extracts from only one of the two hearts, so the conclusion in the results seems overstated. Again, statistics are needed, and control blots to determine whether ItgA5 and ItgA7 are up only in sarcolemma or also in bulk heart tissue. Why do integrins also appear to be up in one of the Thbs3^{-/-} sham controls?

The reviewer probably questions the results in Figure 4 since Figure 3 does not contain data from Thbs3^{-/-} hearts. With respect to Figure 4, we agree that the results presented need revision and

quantitation to make a more conclusive case. Hence we have repeated this experimentation and added quantitation and replaced the blots and added a new Supplemental Figure. We added quantitative statistical analysis that shows no difference in sham operated animals but increased protein levels after TAC (**new Supplemental Figure 4a-f**). We also analyzed integrin protein levels in whole heart tissue that shows no difference in integrin protein levels, which we are showing here, although we mention these results in the revised manuscript (**page 9**).

5. Why were Thbs1/2/3/4/5^{-/-} mice not included in Fig 5f and 5g? That is relevant control to provide a mechanistic correlate for the phenotypes in Fig 5. It is notable that Thbs3^{-/-} show the same inconsistent ItgB1D response in 5g as in Fig 4g.

We agree with the reviewer that Thbs1/2/3/4/5^{-/-} mice are a valuable control for the results presented in Figure 5f and g. As requested we added western blots and immunohistochemical experiments for Thbs1/2/3/4/5^{-/-} hearts to the revised **Figures 5f and 5g**. Quantitative analysis of the western blot experiments revealed significant difference in integrin protein levels comparing Thbs3^{-/-} and Thbs1/2/4/5^{-/-} hearts (**new Supplemental Figure 4g-i**).

6. Fig 7 addresses some of the concerns raised regarding effects of Thbs3 on integrin localization in Fig 3. However, it is unclear whether the modest decreases in surface ItgA5 and ItgB1D are significant relative to the Adbgal control or reproducible. Replicate experiments need to be presented in the supplement and statistical analysis performed.

We agree with the reviewer that presentation of replicate experiments and statistical analysis would be beneficial for the interpretation of the results. We present now 3 replicate experiments showing similar results and quantified the western blot data showing significant changes in Thbs3 overexpressing cells. The replicate experiments are contained within the "**uncropped gel image file**" and the quantitative data are now shown in new **Figure 8**. The quantitation is consistent with the original images we showed, which are now shown in the uncropped gel image file attachment.

7. Given the evidence that exogenous Thbs1 was not effective in earlier experiments, how can the proposed mechanism explain the uptake data in Fig 7f. Does exogenous Thbs3 enhance uptake of surface integrins?

We did not present data analyzing Thbs1 in this manuscript, but maybe the reviewer means Thbs3? Assuming it is Thbs3 the reviewer raises an important question as to if exogenous Thbs3 has an effect on cell surface localization or uptake of integrin proteins. We analyzed the effects of exogenous recombinant Thbs3 and did not observe any changes on cell surface localization or uptake of integrin proteins. We have added these new data to the revised **Figure 8a-h**

8. The reversed activity of AdThbs3RGD in integrin trafficking is interesting, but the integrin localization and uptake data are limited to a RGD-binding integrin. Prior studies indicated that the ItgA7/B1 interaction with its major ligand laminin is not RGD dependent. Do AdThbs3 and AdThbs3RGD have differential effects on surface localization and uptake of ItgA7?

The reviewer raises an important point about the effects of Thbs3 and Thbs3RGD on integrin $\alpha 7$. We agree that our study would greatly benefit from comparing the Thbs3 and Thbs3RGD mediated effects on integrin $\alpha 7$. However, we were unable to detect endogenous integrin $\alpha 7$ in neonatal rat cardiomyocytes. During the neonatal stage a shift in integrin expression occurs (Cardiovasc Res. 2000 Sep;47(4):715-25. Expression of alpha and beta integrins during terminal differentiation of cardiomyocytes. Maitra N et al.). Integrin $\alpha 7$ expression increases after birth which might explain the technical difficulties in detecting this protein in early neonatal cells. While we did assess and quantify $\alpha 7$ integrin content in the sarcolemma of Thbs3 DTG adult hearts and in the hearts of Thbs3^{-/-} mice, and hearts of the 4-way and 5-way KO hearts, which generated a uniform profile of its membrane residence as affected by Thbs3, we did not assess the effect of the RGD mutant on this effect for technical reasons related detection as stated above for the assay we had to use. However, Thbs4 did not cause a clearance of integrins from the cell surface in vitro and in vivo (**Figure 3a and Figure 8a-d**) despite having this RGD domain so we believe the mechanism is slightly more complicated and is related to integrin turnover and intracellular vesicular shuttling in coordination with CRT and CXN, which we are still investigating. This is the only comment from all three reviewers that we could not directly address with new experimental data, so we hope that the reviewer will understand that we are still working on this issue but that it's not straightforward to address from a technical perspective.

Reviewer #3 (Remarks to the Author):

Specific Comments:

The concept of integrin shedding as a component of the remodeling process is appealing, especially in the context of the data presented. It is the opinion of this reviewer that this component, now found in the discussion, could also appear earlier in the manuscript.

We value the reviewers comment and included the concept of integrin shedding and remodeling in the discussion section (page 15).

Electron micrographs to depict increased membrane vesicles in Thbs3 DTG hearts are blurry and not convincing based on the limitations of the small field provided. It also appears that the mitochondria morphology may be altered. Could the authors comment on this?

We thank the reviewer for the constructive concern and we have replaced the electron micrographs in **Figure 1h** to be more clear, and we have added a larger and higher magnification EM image of the Thbs3 DTG heart in the **New Supplemental Figure 1d**. We did not observe any differences in mitochondrial morphology comparing Thbs3 overexpressing hearts with control hearts. The new electron micrograph in **Supplemental Figure 1d** is also labeled to show all the key intracellular structures and organelles.

In general, the rigor of the immunoblotting data on cardiac sarcolemma membrane preps as an approach to show changes in membrane expression of proteins is insufficient in its current execution. This data is central to the argument that Thbs affect cell surface expression of integrins. Quantitation (including n-values) of protein intensities are needed.

The reviewer raises an important point about insufficient rigor of the immunoblotting data on cardiac sarcolemma membrane preps and the statistical quantitation. We have gone back and added more experimentation and performed extensive quantitation throughout our results and we have added new quantitative data (**Figure 1a, Supplementary Figure 1b, Supplementary Figure 3a-k, Supplementary Figure 4a-i, and Figure 8a-h**).

Validation of the membrane enrichment is lacking (are there non-membrane associated proteins in the prep? What is the extent of membrane enrichment? Do the preps contain internal membranes or cell surface membranes?).

We agree with the reviewer that control experiments of the membrane enrichment would be helpful in showing the reviewers that our biochemical purification procedure was robust. We ran such control experiments to validate the extracts used in Figure 3a and Figure 4g. We used actin to show cytoplasm, BiP to show ER content, Gorasp2 to show Golgi and Cacna1c to show sarcolemma. The data attached here show a strong enrichment for the sarcolemma protein fraction in the lanes labeled "membrane" with almost no contamination from the ER, Golgi or cytoplasm. Controls of the supernatant and pellet did show these other proteins from the 3 other compartments, especially in the "supernatant" protein fraction (**see attached below**). These results validate the membrane biochemical purification protocol that was used. Finally, as questioned by **reviewer 2 (response #1 and #4)** we also blotted for total integrin content of $\alpha 5$, $\alpha 7$ and $\beta 1$ from whole cell heart tissue extract and showed that total amounts of these integrins were not changed by Thbs3, validating that only the membrane content of these integrins was

altered such as by shuttling or sheading by Thbs3 or Thbs4 (see data given above to reviewer 2 in responses #1 and #4)

There are many approaches to increase the rigor of these fairly standard biochemical experiments. Standard MW markers are needed.

We added standard molecular weight markers to all Figures.

An additional approach, such as localization of integrins in the heart, should be provided to support that Thbs affects integrin membrane localization. While this is shown for integrin beta1 in figure 5f, the study is limited.

We value the reviewers comment and included immunohistochemical analysis of $\beta 1$ integrin, integrin $\alpha 7$ and $\alpha 5$ integrin in **new Supplemental Figure 3l-n** for tTA control and Thbs3 DTG hearts. The data clearly shows that the Thbs3 transgene down regulates surface antibody reactivity for $\alpha 5$ and $\alpha 7$ integrin, with a mild reduction in $\beta 1$ integrin. While the total levels of these same integrins are not different in whole tissue cytoplasmic extracts from these hearts, in immunohistochemical experiments the antibody more effectively shows areas where these integrins concentrate, such as the plasma membrane, producing a qualitative assessment of plasma membrane content. We also extended **Figure 5f** with integrin $\beta 1$ staining on Thbs1/2/3/4/5^{-/-} sections (**Figure 5f**).

The surface biotinylation of NRVMs treated with Ad expressing Thbs is a valid approach to addressing cell surface expression (Figure 7f). However, the data in its current form isn't convincing.

We agree with the reviewer that the data shown in old Figure 7f could be enhanced. We added two replicate experiments and all the raw gel file images are shown in the "**uncropped gel images**" file, and we have added an entire **new Figure 8** that shows the quantification of these results. Moreover, we also included analysis of recombinant Thbs3 and its effect on integrin cell surface localization, which was unchanged compared to dramatic changes when Thbs3 was expressed intracellularly (**See new Figure 8a-h and file of uncropped gel images**).

Figure 3a: Which laminin polypeptide is shown in the panel? Please use correct symbols for proteins.

We apologize to the reviewer for the unclear labeling of the laminin protein. The correct symbol is laminin2 ($\alpha 2$ chain) and we corrected the figure.

What does the arrow refer to on the panel? Can the author comment on how they conclude that the upper Thbs3 band is non-specific? Were antigen competition studies performed?

The reviewer raises an important point about the specificity of the Thbs3 band in **Figure 3a**. We repeated the immunoblot and were able to obtain a clearer picture of Thbs3 protein with a newly purchased aliquot of the antibody. We validated the antibody with tissue from our Thbs3 KO mice and we know for certainty which is the correct band. However, we have replaced this blot in **Figure 3a** now and removed the arrows because the correct band is completely obvious. This is now consistent across all our western blots for Thbs3 in all our figures.

What is the evidence that the preps represent the cell surface? While Cacna1c serves as a positive control for sarcolemma protein content, a negative marker should be provided (actin or other). Do the preps contain ER/Golgi membranes as well?

We agree with the reviewer that control experiments of the membrane enrichment would be helpful in showing the reviewers that our biochemical purification procedure was robust. We ran such control experiments to validate the extracts used in Figure 3a and Figure 4g. We used actin to show cytoplasm, BiP to show ER content, Gorasp2 to show Golgi and Cacna1c to show sarcolemma. The data attached here show a strong enrichment for the

Figure 3b: The IP experiments lack rigor in their current state. Input microsomal fractions should be shown and reciprocal IPs with Thbs3 antibody showing pull down of integrin beta1 would strengthen the evidence of protein-protein interactions.

We agree with the reviewer that our experiment in Figure 3b is confusing. The experiment presented in Fig3b is an immunoprecipitation of intact microsomes using an integrin $\beta 1$ antibody that recognizes the c-terminal (cytoplasmic) domain of integrin $\beta 1$. This allows the isolation of integrin $\beta 1$ containing vesicles. The experiment shows that Thbs3 localizes to the same vesicles as integrin $\beta 1$, and we do not believe that the reciprocal experiment holds meaning. We apologize to the reviewer for the unclear description of the experiment presented in Fig3b, which we have now revised to make it more clear in the revised results section. We have also added control blots to show the input of these proteins in the given fraction as requested (**new Figure 3b**).

Can the authors comment on the two protein bands observed with the integrin beta1 antibody? The reviewer raises an important point about the two bands detected with the $\beta 1$ integrin antibody. It is common that at least two isoforms of $\beta 1$ integrin are detected, the higher molecular weight band represents the mature form whereas the lower band represents the precursor form of $\beta 1$ integrin (J Clin Invest. 2013 Oct;123(10):4294-308. doi: 10.1172/JCI64216. Integrins protect cardiomyocytes from ischemia/reperfusion injury. Okada H et al. “mature $\beta 1$ D integrin expression (top band 130kDa) ... $\beta 1$ D precursor (immature, bottom band 118kDa)”)

Figure 4g: Are the blots for sham and TAC treated samples from the same exposure? In other words, is it possible to compare the sham and TAC samples? The data supporting differences in sarcolemma localization of integrins upon Thbs3 knockout is weak in its current form and would be strengthened by follow up experiments. These could include localization on cardiac sections using integrin antibodies or cell surface labeling.

We want to thank the reviewer for this valuable comment and we added quantitative data of the immunoblots of Sham and TAC operated animals that now allow the direct comparison of the two panels (see **new Supplementary Figure 4a-f**). We included more immunohistochemical staining of $\beta 1$ integrin in **Figure 5f**, which represents WT, Thbs3^{-/-}, Thbs1/2/4/5^{-/-} and Thbs1/2/3/4/5^{-/-} hearts after 1 week of TAC.

Multiple publications showed upregulation of integrins in different models of cardiac stress: Pressure overload (Histochem Cell Biol. 2002; 118:431–439 DOI 10.1007/s00418-002-0476-1; Modulation of integrins and integrin signaling molecules in the pressure-loaded murine ventricle. Babbitt CJ et al.)

Myocardial infarct (Circulation Research. 1999;85:1046–1055; Remodeling of Cell-Cell and Cell–Extracellular Matrix Interactions at the Border Zone of Rat Myocardial Infarcts; Matsushita T et al.)

Review (Circ Res. 2001;88:1112-1119. Integrins and the Myocardium; Ross RS et al.)

Review (Journal of Signal Transduction; 2011 doi 10.1155/2011/521742; Integrins Are the Necessary Links to Hypertrophic Growth in Cardiomyocytes; Harston RK et al.)

Controls to support the stringency of the sarcolemma enrichment from heart samples is lacking. See our response above to this same concern raised 2x before already, which we have given above in 2 locations, along with control data

What are the multiple bands in integrin beta1D immunoblots?

The reviewer raises an important point about the two bands detected with the β 1 integrin antibody. It is common that at least two isoforms of β 1 integrin are detected, the higher molecular weight band represents the mature form whereas the lower band represents the precursor form of β 1 integrin (J Clin Invest. 2013 Oct;123(10):4294-308. doi: 10.1172/JCI64216. Integrins protect cardiomyocytes from ischemia/reperfusion injury. Okada H et al. “mature β 1D integrin expression (top band 130kDa) ... β 1D precursor (immature, bottom band 118kDa)”)

The data in figure 5f reveal integrin beta1D expression at the sarcolemma in WT and Thbs3 nulls after TAC. There is a notable increase in integrin expression at the sarcolemma in TAC treated THBS3-nulls relative to WT hearts. However, the immunoblot data from Figure 4g suggests that there are no differences in integrin beta1D abundance in microsomal preps from these same genotypes. It is unclear what the differences are between blots shown in Figure 5g and 4g as they both appear to show integrin beta1, integrin alpha 5 and integrin alpha 7 staining for the same TAC treated WT and Thbs3-nulls.

We value the comment of the reviewer and apologize for the unclear representation of the data in Figure 4g and Figure 5g. Figure 4g shows data obtained from experiments after 12 weeks of TAC surgery, which we have tried to better clarify in the revised paper. Figure 5g shows data obtained from experiments after 1 week of TAC surgery since we were unable to obtain living Thbs1/2/4/5-/- animals after 2 weeks TAC (which is shown in Figure 5c).

REVIEWERS' COMMENTS:

Reviewer #1 (Remarks to the Author):

In the revised version, the authors have addressed many of my concerns.

Unfortunately, my major criticism on the questionable pathophysiologic significance of the findings was not adequately addressed. The proposed integrin-dependent mechanism is supported in vivo mostly by overexpression experiments. Although the overexpression models are useful in understanding downstream signaling pathways, they are artificial, and do not provide information on endogenous responses following injury (which is the important information we need to understand cardiac disease).

In the current manuscript the only evidence supporting the proposed integrin-dependent mechanism in a pathophysiologically relevant model of myocardial disease (pressure overload) is associative, showing increased beta1D, alpha5 and alpha7 integrin levels. This is not sufficient to support the notion that this is the only, or the predominant mechanism of TSP3 action that is operative in the remodeling myocardium.

In their response, the authors make 2 points: a) they emphasize the time it took to generate the genetic models analyzed here. The reviewer appreciates the authors' efforts and their careful work. However, the time it takes to do something is not a measure of the significance of the results. b) they highlight the advantages of the comparison between the 5 and 4-way KO mice, animals lacking all TSPs, vs all TSPs except Thbs3 in understanding the role of Thbs3. However, these mice study effects of Thbs3 in an ALTERED environment associated with the complete absence of all TSPs. It is known that TSP1, TSP2 and TSP4 have significant effects in the pressure-overloaded heart. Again this model does not provide any support to the role of the proposed mechanism in the endogenous response to injury (which is the major question).

Despite these concerns, the reviewer appreciates the authors' careful work and feels that this is suitable for a high-quality journal, such as Nature Communications. My recommendation is to tone down the conclusion on the exclusive role of the proposed pathway, acknowledging the significant limitations of the work.

Minor comment:

The authors do not show the data on total integrin expression in the heart ("data not shown"). Considering the unlimited capacity of the supplement, this is not acceptable. Please add this into the supplement.

Reviewer #2 (Remarks to the Author):

This revised manuscript presents novel and convincing genetic evidence that thrombospondin-3 (Thbs3) expression promotes cardiomyopathy, which was unexpected given the opposite activity of thrombospondin-4. The genetic data are strong and utilize both loss and over-expression alone and in the context of deleting other thrombospondin genes. The authors propose a reasonable mechanistic basis for this divergence based on differential regulation of integrin trafficking, although other potential mechanisms cannot be excluded.

The additional controls provided in the revised manuscript address all of this reviewer's major concerns.

Reviewer #3 (Remarks to the Author):

The authors very rigorously and thoroughly addressed all of the comments with many additional experiments and controls. In fact, this may be the most deeply considered and responsive rebuttal that I have reviewed. I also considered the comments from Reviewer #1 regarding the non-physiological model systems, which is a major critique of the manuscript. I understand the reviewer's viewpoint here. I also appreciate the author's multiyear efforts to generate the multiple knockout and transgenic mice, but don't feel that intensive effort alone is justification for publication. However, I do strongly feel that the authors' scientific approach is highly valid to address the biological questions set forth by the authors and is revealing of thrombospondin and integrin function.

RESPONSE TO REVIEWERS' COMMENTS:

Reviewer #1 (Remarks to the Author):

In their response, the authors make 2 points: a) they emphasize the time it took to generate the genetic models analyzed here. The reviewer appreciates the authors' efforts and their careful work. However, the time it takes to do something is not a measure of the significance of the results. b) they highlight the advantages of the comparison between the 5 and 4-way KO mice, animals lacking all TSPs, vs all TSPs except Thbs3 in understanding the role of Thbs3. However, these mice study effects of Thbs3 in an ALTERED environment associated with the complete absence of all TSPs. It is known that TSP1, TSP2 and TSP4 have significant effects in the pressure-overloaded heart. Again this model does not provide any support to the role of the proposed mechanism in the endogenous response to injury (which is the major question). Despite these concerns, the reviewer appreciates the authors' careful work and feels that this is suitable for a high-quality journal, such as Nature Communications. My recommendation is to tone down the conclusion on the exclusive role of the proposed pathway, acknowledging the significant limitations of the work.

As suggested we have toned down our conclusions throughout the revised manuscript and on page 13 of the discussion we have qualified the physiologic significance of the 4-way KO mice, bringing up the qualification that this model is somewhat unphysiologic itself as during injury all 5 Thbs genes appear to go up in the heart (see page 13)

Minor comment:

The authors do not show the data on total integrin expression in the heart ("data not shown"). Considering the unlimited capacity of the supplement, this is not acceptable. Please add this into the supplement.

We have revised the "data not shown" demarcations throughout the revised manuscript and in this particular instance we have removed mention of these data that were not shown

Reviewer #2 (Remarks to the Author):

This revised manuscript presents novel and convincing genetic evidence that thrombospondin-3 (Thbs3) expression promotes cardiomyopathy, which was unexpected given the opposite activity of thrombospondin-4. The genetic data are strong and utilize both loss and over-expression alone and in the context of deleting other thrombospondin genes. The authors propose a reasonable mechanistic basis for this divergence based on differential regulation of integrin trafficking, although other potential mechanisms cannot be excluded.

The additional controls provided in the revised manuscript address all of this reviewer's major concerns.

OK thank you!

Reviewer #3 (Remarks to the Author):

The authors very rigorously and thoroughly addressed all of the comments with many additional experiments and controls. In fact, this may be the most deeply considered and responsive rebuttal that I have reviewed. I also considered the comments from Reviewer #1 regarding the non-physiological model systems, which is a major critique of the manuscript. I understand the reviewer's viewpoint here. I also appreciate the author's multiyear efforts to generate the

multiple knockout and transgenic mice, but don't feel that intensive effort alone is justification for publication. However, I do strongly feel that the authors' scientific approach is highly valid to address the biological questions set forth by the authors and is revealing of thrombospondin and integrin function.

We did qualify the physiologic significance of our data with respect to the overexpression model and the 4-way KO mice on page 13 of the discussion.